# UPF3A is dispensable for nonsense-mediated mRNA decay in mouse pluripotent and somatic cells

Chengyan Chen[1,*], Yanmin Shen[1,*], Luqian Li[1], Yaoxin Ren[1], Zhao-Qi Wang[3], Tangliang Li[1,2]

**Nonsense-mediated mRNA decay (NMD) is a highly conserved regulatory mechanism of post-transcriptional gene expression in eukaryotic cells. NMD plays essential roles in mRNA quality and quantity control and thus safeguards multiple biological processes including embryonic stem cell differentiation and organogenesis. UPF3A and UPF3B in vertebrate species, originated from a single *UPF3* gene in yeast, are key factors in the NMD machinery. Although UPF3B is a well-recognized weak NMD-promoting factor, whether UPF3A functions in promoting or suppressing NMD is under debate. In this study, we generated a *Upf3a* conditional knockout mouse strain and established multiple lines of embryonic stem cells and somatic cells without UPF3A. Through extensive analysis on the expressions of 33 NMD targets, we found UPF3A neither represses NMD in mouse embryonic stem cells, somatic cells, nor in major organs including the liver, spleen, and thymus. Our study reinforces that UPF3A is dispensable for NMD when UPF3B is present. Furthermore, UPF3A may weakly and selectively promote NMD in certain murine organs.**

## Introduction

In eukaryotic cells, the transmission of genetic information from DNA to proteins is under stringent regulations at multiple layers. Nonsense-mediated mRNA decay (NMD) is a highly conserved gene expression regulation mechanism. NMD surveillances transcriptome quality by eliminating mRNAs containing premature termination codons (PTCs) and thus prevents accumulation of N-terminal truncated protein. Meanwhile, through recognizing other NMD features, such as long 3′ UTRs and 5′ uORFs, NMD regulates stability of around 3–10% of normal RNA transcripts and participates in fine-tuning gene expression (Lykke-Andersen & Jensen, 2015; Kurosaki et al, 2019). Thus, NMD plays important roles in cell fitness, stress response, etc. (Hug et al, 2016). NMD is essential for embryonic development and tissue/organ maintenance (Han et al, 2018).

UPF3A and UPF3B are unique NMD factors among all components of mammalian NMD machinery. UPF3A and UPF3B are two paralogs of Upf3 found in the baker's yeast (Lykke-Andersen et al, 2000). Phylogenetic analysis indicates UPF3A and UPF3B could be generated by a gene duplication event during the emergence of vertebrate species (Shum et al, 2016). UPF3B is a widely accepted mild NMD factor negatively regulating mRNA stability (Lykke-Andersen et al, 2000; Kunz et al, 2006; Chan et al, 2007, 2009). Furthermore, UPF3B participates in early and late translation termination, suggesting orchestrated roles of UPF3B in the life regulation of RNA turnover and protein synthesis (Gao &Wilkinson, 2017; Neu-Yilik et al, 2017). Interestingly, although structural and biochemical analyses show that UPF3B is one of the central factors in NMD, UPF3B loss causes very mild or even negligible NMD defects because UPF3B only regulates the stability of a small proportion of RNA targets (Tarpey et al, 2007; Huang et al, 2018a). Meanwhile, UPF3A has weak NMD-promoting activity in vitro (Lykke-Andersen et al, 2000); in UPF3B-deficient mammalian cells, UPF3A is considered to compensate for UPF3B loss (Chan et al, 2007, 2009; Wallmeroth et al, 2022; Yi et al, 2022) because UPF3A and UPF3B, through their middle domain, could bind to the MIF4GIII domain of NMD factor UPF2 (Bufton et al, 2022).

Not like other NMD factors, the knockdown of Upf3a, Upf3b, or both shows no obvious developmental defects in zebrafish (Wittkopp et al, 2009). Upf3a or Upf3b KO zebrafishes are all viable (Ma et al, 2019). In mammals, KO mice of *Smg1, Upf1, Upf2,* or *Smg6* are embryonic lethal (Han et al, 2018). *Upf3b* null mice generated by the gene trapping strategy are viable and have very mild neurological symptoms (Huang et al, 2018b). Interestingly, *Upf3a* KO mice are early embryonic lethal (Shum et al, 2016), suggesting UPF3A may have different roles in NMD. A detailed analysis of UPF3A functions in HEK293 cells, Hela cells, mouse pluripotent cells (P19: embryonic carcinoma cell), somatic cells, and major organs, such as olfactory bulbs and testes, identified a novel function of UPF3A as a general and strong NMD repressor in mammals (Shum et al, 2016).

Recently, two back-to-back articles from Gehring's and Singh's groups revisited the NMD functions of UPF3A and UPF3B with human cell lines (Wallmeroth et al, 2022; Yi et al, 2022). Using

[1]State Key Laboratory of Microbial Technology, Shandong University, Qingdao, China [2]School of Basic Medical Sciences, Hangzhou Normal University, Hangzhou, China [3]Leibniz Institute on Aging - Fritz Lipmann Institute, Jena, Germany

Correspondence: li.tangliang@sdu.edu.cn
*Chengyan Chen and Yanmin Shen contributed equally to this work

transcriptomic data and qPCR analysis on the expressions of known NMD target genes including *RSRC2*, *SRSF2*, and *ZFAS1*, Wallmeroth et al found that overexpression of UPF3A or UPF3A knockdown in HEK293 cells and Hela cells does not affect NMD efficiency (Wallmeroth et al, 2022). Furthermore, NMD in UPF3B KO cells is generally functional, and co-depletion of UPF3A and UPF3B results in stronger NMD inhibition, as revealed with RNA-Seq and qPCRs. These data indicate that UPF3 paralogs in humans are functionally redundant and could compensate for each other in NMD when one of the UPF3 paralogs is missing. Interestingly, Yi et al applied human colorectal carcinoma HCT116 cell line with a near-diploid genome carrying only one UPF3B copy and generated two independent UPF3B KO cell lines with the CRISPR-Cas9 technology (Yi et al, 2022). They found that two UPF3B KO HCT116 cell lines showed mild NMD defects because qPCR analysis revealed that PTC-containing iso-forms of *ILK*, *NFKBIB*, *RPS9*, and *SRSF3* are unregulated. The mild NMD deficiency is further supported by the finding that RNA-seq analysis showed an accumulation of PTC-containing isoforms in the transcriptome from UPF3B KO cells. Furthermore, Yi et al showed knockdown of UPF3A in UPF3B KO cells further enhanced NMD inhibition. Intriguingly, UPF3A KO HCT116 cells have a minimal effect on the abundance of PTC-containing isoforms. These two findings from Gehring's and Singh's groups reconcile a conclusion that, in human cells, UPF3A does not repress NMD, but is dispensable for NMD. Furthermore, UPF3A protein is up-regulated and promotes NMD when UPF3B protein is depleted (Wallmeroth et al, 2022; Yi et al, 2022).

To briefly summarize the abovementioned findings, research works conducted with human samples strongly indicate that UPF3A is dispensable for NMD (Lykke-Andersen et al, 2000; Kunz et al, 2006; Chan et al, 2007, 2009; Wallmeroth et al, 2022; Yi et al, 2022), whereas Shum et al found that UPF3A, in general, represses NMD in mouse cells and tissues (Shum et al, 2016). NMD may have species or cell–type specificity. Thus, we set out to reinvestigate the UPF3A's function with mouse cells and tissues. To this end, we generated a *Upf3a* conditional KO mouse by introducing two loxP sites floxing *Upf3a* exon 3, which is identical to the published *Upf3a* conditional gene targeting strategy (Shum et al, 2016). We produced a panel of *Upf3a*-deficient embryonic stem cells (ESCs), somatic cells, and tissues. To our surprise, with extensive qPCR and semi-quantitative RT–PCR analysis on the expressions of more than 30 NMD targets in ESCs, somatic cells, and various tissues including liver, spleen, and thymus, we found that *Upf3a* deficiency does not cause changes on the expressions of NMD targets in these mouse cells and tissues; thus, UPF3A does not play a role as a NMD repressor. Our results reinforce the conclusion that UPF3A is dispensable for mammalian NMD when UPF3B exists (Wallmeroth et al, 2022; Yi et al, 2022).

# Results

## Characterizing an antibody capable to detect mouse endogenous UPF3A and UPF3B proteins

Before our analysis on UPF3A's function in NMD, we validated a commercial antibody (Abcam 269998) that is claimed to detect

UPF3A and UPF3B simultaneously in a single blot. First, we cloned mouse *Upf3a* and *Upf3b* cDNAs into the pEGFP-C1-EF1A vector (Li et al, 2015) and conducted Western blot with protein samples isolated from U2OS cells transiently transfected with GFP empty vector, GFP-mUpf3a, or GFP-mUpf3b constructs. We could detect specific protein bands approximately at sizes of predicted GFP-mUpf3a and GFP-mUpf3b in protein lysates from U2OS cells transfected with GFP-mUpf3a and GFP-mUpf3b, respectively (Fig S1A and B), indicating that this Abcam UPF3A+UPF3B antibody could be used to detect GFP tagged mouse UPF3A and UPF3B proteins.

To further investigate whether this antibody is capable to detect endogenous UPF3A and UPF3B, we used siRNA to knockdown endogenous UPF3A and UPF3B in mouse ESCs. Through Western blot, we found in control samples (untreated, reagent treated, and non-targeting siRNA treated samples), hybridization of the Abcam UPF3A+UPF3B antibody revealed two distinct bands between 52 and 66 kD. siRNA-Upf3b–treated samples exhibited diminishment of the upper bands and resulted in the great induction of lower bands (Fig S1C). This finding strongly suggested that the upper band corresponds to UPF3B and the lower band represents UPF3A because previous studies all showed that UPF3B depletion by gene KO or knockdown strategies could dramatically increase UPF3A protein level (Chan et al, 2009; Shum et al, 2016; Wallmeroth et al, 2022; Yi et al, 2022). The fact that the lower band corresponds to UPF3A is further supported by the Western blot analysis on ESC treatment with three *Upf3a* siRNAs. We found in siRNA-Upf3a–treated ESC samples, UPF3A (lower band) are all mildly decreased. Thus, Abcam UPF3A+UPF3B antibody is reliable in detecting endogenous mouse UPF3A and UPF3B proteins.

## UPF3A is dispensable for NMD in mouse ESCs

To override the embryonic lethality of *Upf3a* null (Shum et al, 2016), we generated a *Upf3a* conditional KO mouse (*Upf3a*^flox/flox: *Upf3a*^f/f) with CRISPR-Cas9 technology (see "the Materials and Methods section" for details, Fig 1A) and crossed *Upf3a*^f/f mouse with Cre-ER^T2+ mouse, an inducible Cre transgenic line (Fig 1B). Through intercrossing *Upf3a*^f/f Cre-ER^T2+ mouse with *Upf3a*^f/f mouse, we isolated E3.5 blastocysts and established several *Upf3a*–inducible deletion ESC lines. Four female ESC lines showing the typical ESC morphology were selected for further analysis. We treated these ESC lines with 4-OHT for 5 d and confirmed *Upf3a* exon 3 is completely deleted with normal genotyping and qPCR (Figs 1C and 2A). Interestingly, qPCR with primer pairs specifically detecting exons 1–2, exons 7–8, or exons 8–9 showed that 4-OHT–treated ESCs have around 80% reduced mRNA levels of *Upf3a*, which could be caused by an efficient NMD of truncated *Upf3a* mRNAs because *Upf3a* exon 3 deletion will generate a PTC (Fig S2).

To further substantiate the successful establishment of UPF3A KO ESCs, we conducted Western blot with aforementioned UPF3A+UPF3B antibody and found UPF3A proteins are completely absent in 4-OHT treatment *Upf3a*^f/f Cre-ER^T2+ ESC lines, indicating *Upf3a* KO ESC lines (designated as *Upf3a*^△/△ ESCs or UPF3A KO mESCs) are established (Figs 1D and S3A). UPF3A KO mESCs are all viable and proliferative normally (Figs 1E and S3B). They show no phenotypic difference with their parental clones (Fig 1E, left panel).

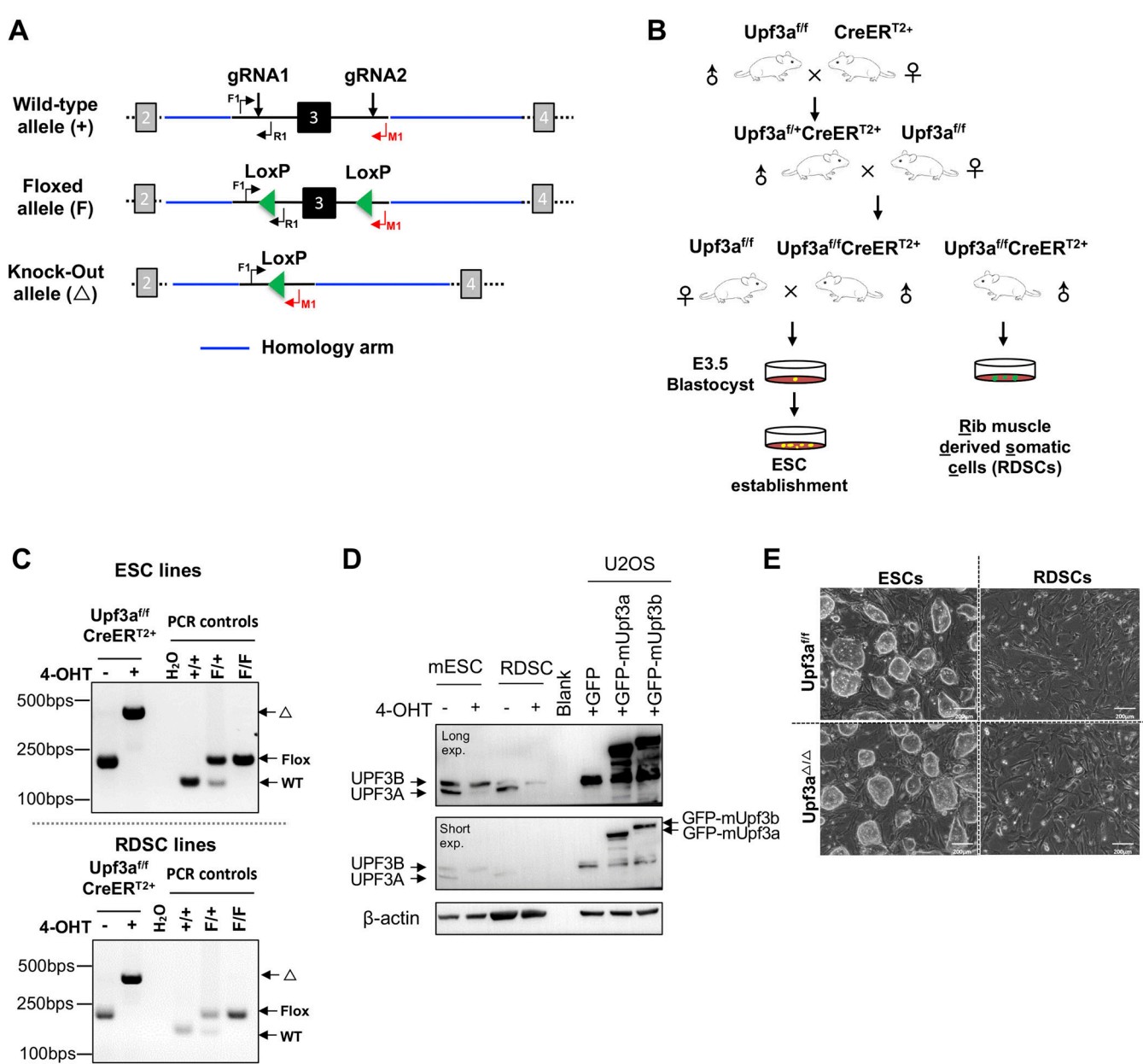

**Figure 1. Generation of *Upf3a* KO embryonic stem cells (ESCs) and somatic cells.**
**(A)** Strategy to generate *Upf3a* conditional KO mouse. Exon 3 of *Upf3a* is chosen to be conditionally deleted by Cre recombinase in vitro and in vivo. Wt allele, floxed allele (F), and KO allele (△) are shown. The locations of genotyping primers (F1, R1, and M1) are marked on these *Upf3a* alleles. **(B)** Mating strategy to generate *Upf3a*-inducible KO mouse line (*Upf3a*^f/f Cre-ER^T2+). *Upf3a*^f/f Cre-ER^T2+ mice are further used to produce *Upf3a*-inducible KO embryonic stem cells and somatic cells from rib muscle (RDSCs). **(C)** PCR analysis on *Upf3a* locus (exon 3) deletion in *Upf3a*^f/f Cre-ER^T2+ ESCs and RDSCs after 4-OHT treatment. WT allele, floxed allele (F), and KO allele (△) are marked. **(D)** Western blot analysis on UPF3A and UPF3B protein expressions in *Upf3a*^f/f Cre-ER^T2+ ESCs and RDSCs after 4-OHT treatment. Protein lysates from U2OS cells expressing GFP-mUpf3a and GFP-mUpf3b are used to validate the UPF3A+UPF3B antibody. β-Actin is used as a loading control for Western blot. **(E)** Representative images of control ESCs/RDSCs (*Upf3a*^f/f) and *Upf3a* KO ESCs/RDSCs (*Upf3a*^△/△).
Source data are available for this figure.

To investigate UPF3A's role in NMD, we conducted qPCR analysis on mRNA transcripts from the following sets of NMD target genes (the features of NMD targets selected for this study are summarized in Fig S4): Genes including *Cdh11, Ire1, Smad5, Smad7, Gas5,* and *Snord22* were previously used to characterize UPF3A as an NMD repressor in mouse (Shum et al, 2016); genes including *Atf4, Ddit3, 1810032O08Rik, Smg5, Smg6, Auf1* (PTC specific), *Hnrnpl* (PTC specific) were widely used or validated as NMD targets in mice (Weischenfeldt et al, 2008, 2012; Li et al, 2015); *Eif4a2* (PTC specific) was recently identified in *Smg5, Smg6,* and *Smg7* KO ESCs generated with the CRISPR-Cas9 technology (Huth et al, 2022); *Upf3b, Smg7, Eif4a1* (PTC specific), *Mettl23* (PTC specific), *Sfrs10* (PTC specific), *Snrpb* (PTC specific), *Hnrnpa2b1* (PTC specific), *Luc7l* (PTC specific), and *Pkm2* (PTC specific) are designed in the current study.

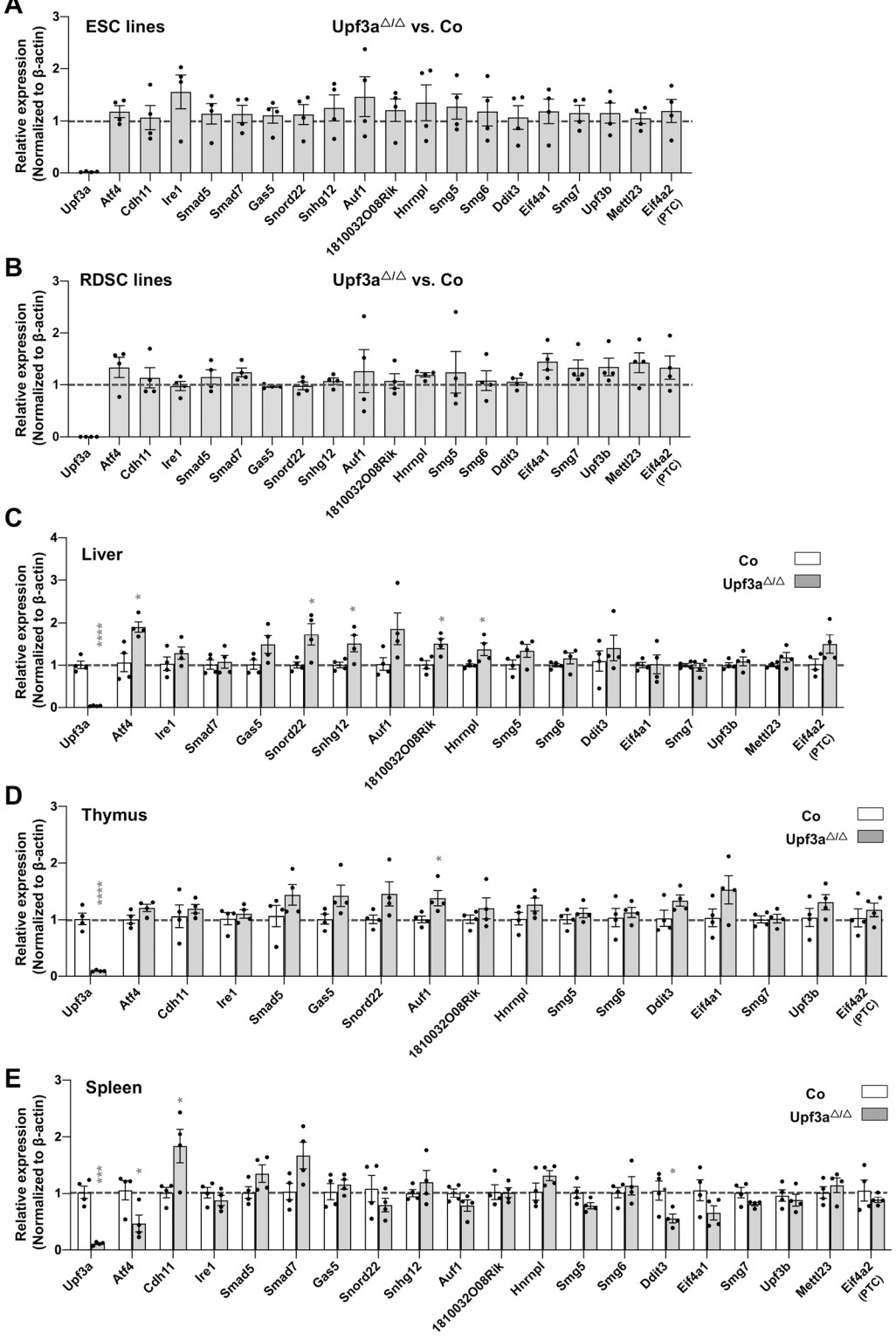

**Figure 2. UPF3A does not repress NMD targets expression.**
**(A, B)** qPCR analysis on expressions of NMD targets in mESCs (A) and RDSCs (B). **(A, B)** Because of considerable variances on expression levels of these NMD target genes between individual ESC and RDSC line, each data point (A, B) represents the relative expression of indicated gene in *Upf3a* KO related to its parental cell line. **(C, D, E)** qPCR analysis on expressions of NMD targets in liver (C), thymus (D), and spleen (E) from control (Co: *Upf3a*$^{f/f}$ Cre-ER$^{T2-}$ + TAM) and *Upf3a* KO (*Upf3a*$^{\triangle/\triangle}$: *Upf3a*$^{f/f}$ Cre-ER$^{T2+}$ + TAM) mice. Note: *, $P < 0.05$; ***, $P < 0.001$; Unpaired $t$ test is used.

In our analysis, previously generated *Smg6* KO ESC was used as a positive control of NMD inhibition (Li et al, 2015) (Figs S5A and S6A). In contrast with strong increases of NMD target transcripts in *Smg6* KO ESCs (Figs S5A and S6A), *Upf3a* KO ESCs showed no obvious difference in transcript levels of all NMD target genes tested (Figs 2A and S6B). Furthermore, we conducted RT–PCR analysis on AS-NMD–generated PTC⁺ transcript accumulation in *Upf3a* KO ESCs. We found a strong enrichment of PTC⁺ isoforms in *Smg6* KO ESCs, but no

difference was detected between control and *Upf3a* KO ESCs (Fig 3A and B). Thus, UPF3A is dispensable for NMD in mouse ESCs when UPF3B is present.

### Knockdown of UPF3B in UPF3A-deficient ESCs inhibits NMD

In human cells, UPF3A and UPF3B could compensate for each other in NMD when one of the paralogs is depleted (Chan et al, 2009; Wallmeroth et al, 2022; Yi et al, 2022). To test this hypothesis in mouse cells, we used siRNAs to knockdown *Upf3b* in our control and *Upf3a* KO ESCs. Semi-quantitative RT–PCR showed that, in control ESCs, knockdown of *Upf3b* has almost no effect on expressions of the PTC⁺ isoforms of *Hnrnpa2b1*, *Luc7l*, *Pkm2*, *Ptbp2*, *Flot1*, and *Sf1* (Fig S7). In *Upf3a* KO ESCs, depletion of UPF3B causes a prominent increasement of PTC⁺ isoforms of these genes. Thus, in mice, UPF3B is responsible for efficient NMD when UPF3A is lost.

### UPF3A is dispensable for NMD in mouse somatic cells

To rule out a possible role of pluripotency on NMD activity, we generated somatic cell lines derived from rib tissues of 4 male *Upf3a*^f/f Cre-ER^T2+ mice. In this study, we named these cell lines as RDSCs (Rib muscle derived somatic cells). qPCR and Western blot analysis showed that 5 d of 4-OHT treatment successfully depleted UPF3A in RDSCs (Fig 1C, lower panel; Figs 1D, 2B, and S3A). Loss of UPF3A does not cause visible cell viability changes (Fig 1E, right panel). We then used qPCR and analyzed expressions of NMD target gene transcripts in control and *Upf3a* KO RDSCs. Previously established *Smg6* KO fibroblast was used as a positive control of NMD inhibition in somatic cells (Figs S5B and S6A) (Li et al, 2015). We found *Upf3a* KO RDSCs, as compared with their parent cell lines, showed no obvious difference in transcript levels of all NMD target genes tested (Figs 2B and S6B). Furthermore, RT–PCR analysis on PTC⁺ isoforms showed *Smg6* KO fibroblasts had a strong accumulation of PTC⁺ isoforms. However, *Upf3a* KO did not result in any detectable change in the PTC⁺ isoforms tested (Fig 3C and D). Thus, in mouse somatic cells, such as fibroblasts generated from rib muscles, UPF3A is dispensable for NMD when UPF3B is present.

### UPF3A is dispensable for NMD in murine tissues

Because NMD may have tissue and cell type specificity, to expand our analysis to other cell types, we used *Upf3a*-inducible KO mice (*Upf3a*^f/f Cre-ER^T2+ mice) and treated them with tamoxifen to induce UPF3A deletion. After another 3 wk, PCR analysis and Western blot show that UPF3A is efficiently deleted in the liver, kidney, and hematopoietic system, including the spleen, thymus, and peripheral blood cells from *Upf3a*^f/f Cre-ER^T2+ mice treated with tamoxifen (Fig S8A and B). These mice are viable and have no visible behavior changes as compared with control animals (*Upf3a*^f/f Cre-ER^T2− mice with tamoxifen injection) (data not shown). We isolated RNAs from livers, spleens, and thymus and used qPCR to determine NMD activity in samples from 4 controls (*Upf3a*^f/f Cre-ER^T2− mice with TAM injection) and four *Upf3a* KO mice (*Upf3a*^f/f Cre-ER^T2+ mice treated with TAM) (Figs 2C–E and S6C). In general, expressions of these NMD targets show no or minimal variances between controls and UPF3A KO samples. In *Upf3a* KO livers, RNA transcripts of well-conserved

NMD targets, such as *Atf4*, *Snord22*, *Snhg12*, *1810032O08Rik*, and *Hnrnpl*, are mildly but all significantly up-regulated, whereas other gene transcripts, such as *Gas5*, *Auf1*, and *Eif4a2* have a trend of increasement (Fig 2C). In *Upf3a* KO thymuses, only transcripts of *Auf1* are significantly increased, whereas transcripts of *Smad5*, *Gas5*, *Snord22*, *Hnrnpl*, *Ddit3*, and *Eif4a1* trend to accumulate (Fig 2D). It is interesting to note, in *Upf3a* KO livers and thymuses, none of the genes tested show reduction at the transcript level (Fig 2C and D). Of note, the expressions of several NMD target genes, such as *Smad5* and *Cdh11* in livers, *Smad7*, *Snhg12* and *Mettl23* in thymuses, could not be quantified because of the technical reason that melting curves of PCR amplicons showed multiple peaks.

In *Upf3a* KO spleens, expression of *Cdh11* is significantly up-regulated, whereas UPR factors, including *Atf4 and Ddit3*, are significantly down-regulated. Expression levels of other 16 NMD targets have no difference between the control and *Upf3a* KO samples. Overall, in spleens, UPF3A does not repress NMD (Fig 2E).

Next, we conducted RT–PCR analysis on AS-NMD–generated PTC⁺ transcripts accumulation in *Upf3a* KO livers and spleens; we found no obvious change in PTC⁺ isoforms expression between control and *Upf3a* KO samples (Fig S9).

## Discussion

UPF3 is widely considered as one of the core factors of NMD machinery, which is a highly conserved mRNA surveillance mechanism in eukaryotes cells (Lykke-Andersen et al, 2000; Kunz et al, 2006; Lykke-Andersen & Jensen, 2015; Hug et al, 2016; Yi et al, 2021). In yeasts and worms, only one *Upf3* locus is identified (Hug et al, 2016). In vertebrate animals, two paralogs of UPF3, that is, UPF3A and UPF3B, exist (Lykke-Andersen et al, 2000; Serin et al, 2001). It is even more intriguing that *the Upf3b* gene is localized on the X chromosomes in mice and humans. Interestingly, although KOs or knockdowns of NMD factors, such as *Upf1*, *Upf2*, *Smg1*, *Smg5*, *Smg6,* and *Smg7*, in various types of cells manifest strong NMD defects (Li et al, 2015; Huth et al, 2022), *Upf3b* knockdowns or KOs have mild or even negligible effects on NMD (Lykke-Andersen et al, 2000; Chan et al, 2007; Huang et al, 2011). Through generating a conditional KO mouse model of *Upf3a* with a traditional gene targeting strategy, Shum et al found UPF3A represses NMD in mouse pluripotent cells (P19 embryonic carcinoma cell), tissue-specific stem cells (neural stem cells and olfactory sensory neuronal precursors), somatic cells (mouse embryonic fibroblasts and neurons), and human 293T cells. NMD targets, such as *Cdh11*, *Smad5*, and *Smad7*, are all down-regulated upon *Upf3a* knockdown (Shum et al, 2016). Meanwhile, mRNAs transcripts of *Atf4*, *Gas5*, and *Snord22* showed tissue-specific NMD activation or repression upon *Upf3a* knockdown (Shum et al, 2016). Because of the strong NMD repression upon *Upf3a* deficiency, Shum et al found *Upf3a* KO mice are embryonic lethal.

Two recent studies revisited the function of UPF3A in humans (Wallmeroth et al, 2022; Yi et al, 2022). Wallmeroth et al used the human HEK293 and Hela cell lines and found UPF3A overexpression or KO does not affect NMD efficiency. Furthermore, Yi et al used the human HCT116 cell line and found that UPF3A functions as a weak

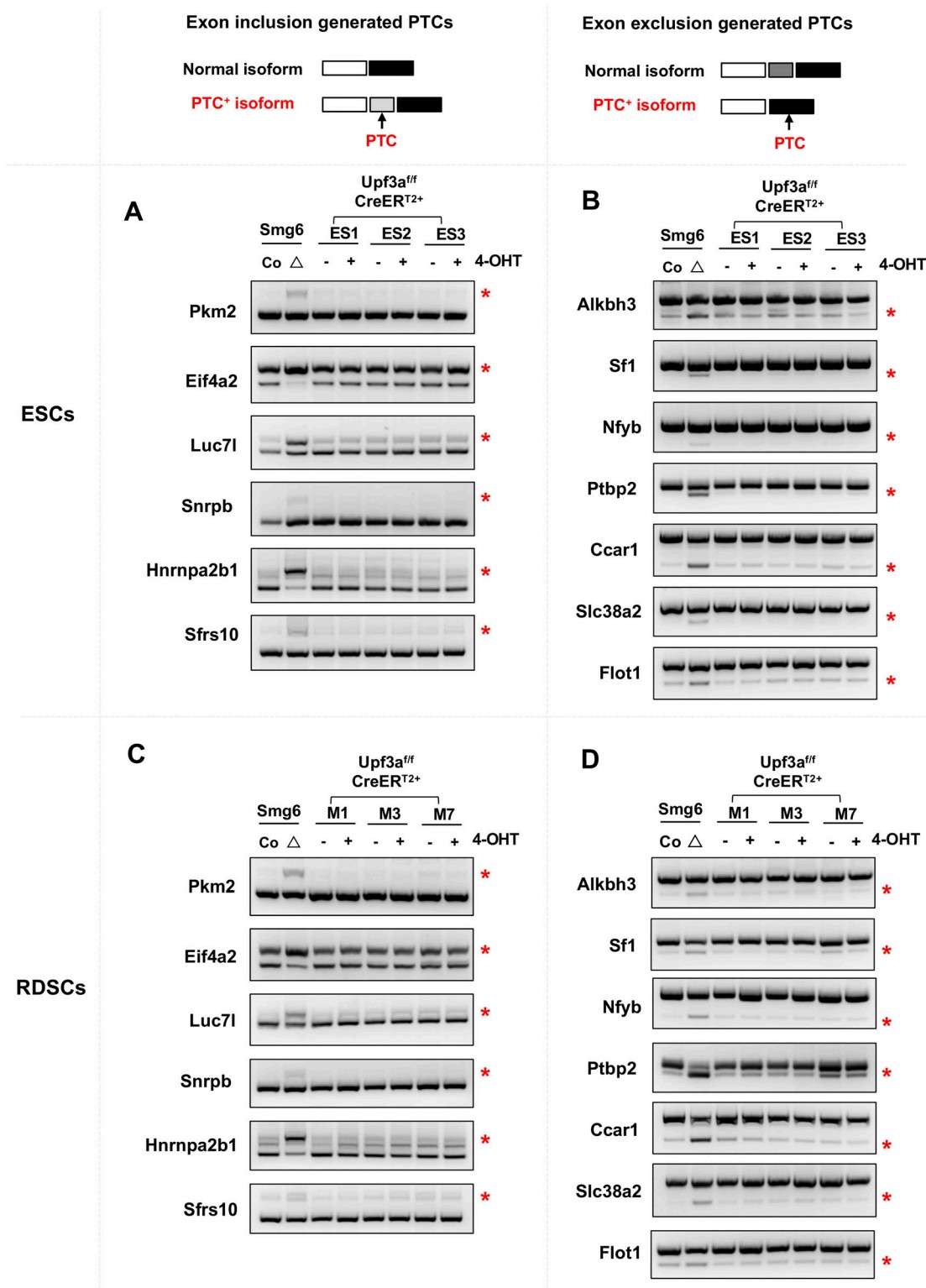

**Figure 3. UPF3A deficiency does not affect AS-NMD in ESCs and RDSCs.**
**(A, B, C, D)** RT–PCR analysis on AS-NMD in *Upf3a* KO ESCs (A, B) and RDSCs (C, D). **(A, B, C, D)** Accumulations of PTC⁺ isoforms generated with exon inclusion events (A, C) and exon skipping events (B, D) are analyzed by RT–PCR. ESC lines (ES1, ES2, ES3) and RDSC lines (M1, M3, M7) are used. AS-NMD defects in *Smg6* KO ESCs and fibroblasts (Smg6△) are used as a positive control of NMD inhibition for this analysis.
Source data are available for this figure.

NMD activator (Yi et al, 2022). UPF3A could compensate for UPF3B loss and function in NMD activation because UPF3A and UPF3B double KO cells have profound NMD defects (Wallmeroth et al, 2022; Yi et al, 2022). These two new findings, together with most of previous results, strongly indicated that UPF3A and UPF3B are mainly weak NMD activators and function redundantly in the human NMD (Lykke-Andersen et al, 2000; Kunz et al, 2006; Chan et al, 2007, 2009). In the discussion parts from these two studies, the authors raised several possibilities on the discrepant UPF3A's roles in the mammalian NMD: (1) UPF3A may function distinctly in mice and humans; (2) UPF3A's role in NMD may have cell-type specificity (Wallmeroth et al, 2022; Yi et al, 2022).

In this study, we used CRISPR-Cas9 technology and adopted the identical gene targeting strategy to generate a new *Upf3a* conditional KO mouse strain. Furthermore, we characterized a newly released rabbit monoclonal antibody which could detect the expression of mouse UPF3A and UPF3B in a single immunoblotting (Fig S1). With these key materials, we successfully produced and characterized *Upf3a* KO mESCs and somatic cells. We found that UPF3A and UPF3B protein expressions are higher in female ESCs than in male somatic cells (mouse rib muscle derived somatic cells) (Fig 1D), indicating that expressions of UPF3A and UPF3B proteins are developmentally regulated. Because undifferentiated female ESCs have two active *Upf3b* gene copies, whereas only one *Upf3b* gene locus is present in male RDSCs, the relative amount of UPF3B versus UPF3A is reduced in male RDSCs.

In our study, we found that deletion of UPF3A was compatible with the life of ESCs and somatic cells, which was similar to UPF3B null mouse cells (Huang et al, 2011). Through qPCR and semiquantitative RT–PCR assays, we found UPF3A loss did not cause decrease in mRNA transcripts of well-recognized NMD target genes. On the contrary, in *Upf3a* KO ESCs, somatic cells, and in various mouse tissues, including the liver and thymus, transcript levels of these well-documented NMD targets remained unchanged or slightly up-regulated. Of note, in *Upf3a* KO livers, mRNA transcripts of NMD target genes, including *Atf4, Snord22, Snhg12, 1810032O08Rik, and Hnrnpl* were mildly but significantly up-regulated, indicating that UPF3A has tissue specificity in NMD. Thus, our study indicates that UPF3A is not a repressor of NMD. Our study, together with most of the previous studies on UPF3A and UPF3B indicates that UPF3 paralogs are weak NMD activators in mammalian cells (Lykke-Andersen et al, 2000; Kunz et al, 2006; Chan et al, 2007, 2009; Wallmeroth et al, 2022; Yi et al, 2022). Furthermore, knockdown of UPF3B in UPF3A-deficient ESCs causes evident NMD inhibition, which is in agreement with the previous hypothesis that UPF3B compensates for UPF3A loss and participates in NMD in mammalian cells.

Of note, in current study conducted with murine cells and tissues, to investigate the NMD efficacy, we only used qPCR and semiquantitative RT–PCR assays and analyzed the expressions of 33 NMD targets with features of PTCs, uORFs, or long 3′ UTRs, which reveals UPF3A is dispensable for NMD in murine cells and tissues. However, we could not rule out the possibility that UPF3A may repress some uncharacterized gene transcripts in certain unexplored mouse tissues, such as the testis with the highest UPF3A expression (Tarpey et al, 2007; Shum et al, 2016). Future transcriptome-wide analysis in *Upf3a*-deficient murine cells or tissues would resolve UPF3A's role in NMD.

# Materials and Methods

## Mice and genotyping strategies

The *Upf3a* conditional KO mouse (*Upf3a*^f/f) was generated by CRISPR-Cas9 gene editing in Cyagen. Cas9 protein, two gRNAs (gRNA1: AAATCTGTTGTTCGTACAGA, gRNA2: CTTGTTACAAGCTTTAGCCG), and a donor vector containing the two loxP sequences in intron 2 and intron 3 of mouse *Upf3a* gene were injected into mouse fertilized eggs. The embryos were transferred to recipient female mice to obtain F0 mice. For validating the two loxP site insertions, PCR products using two pairs of primers (F1: AAAGAACAGTGTGCAATTACTCGG, R1: TTCACAGGTAGGAACGATTCCATT; F2: TGTCCATTTTACCTATCCATTCG, R2: GAGCACTGCGCTACCACCTGACC) were sequenced. For routine genotyping of *Upf3a* alleles, three primers were used: F1, AAAGAACAGTGTGCAATTACTCGG; R1, TTCACAGGTAGGAACGATTCCATT; M1, AGCTCTTACTCTTGAGCCCAC (WT allele: 133 bps; floxed allele: 200 bps; KO allele: 412 bps) (Fig 1A). To generate the *Upf3a*-inducible KO mouse, *Upf3a*^f/f mouse was crossed with Cre-ER^T2+ transgenic mouse line. For genotyping of Cre-ER^T2+ transgene, primers (Cre-ErF: ATACCGGAGATCATGCAAGC; Cre-ErR: GATCTCCACCATGCCCTCTA) were used (Cre-ER^T2+ transgene: 552 bps).

To delete *Upf3a* in adult *Upf3a*^f/f Cre-ER^T2+ mice (age: 6 wk), tamoxifen (T5648; Sigma-Aldrich) was intraperitoneally injected at a dose of 75 mg/kg for three consecutive days. 3 wk after the last tamoxifen injection, mice tissues were collected and processed for Western blot and qPCR analysis.

All animals were maintained under specific pathogen-free conditions at the animal facility of the Shandong University, Qingdao, PR China. Animal care and experiments were performed in accordance with the ethic committee guideline.

## Generation of UPF3A-inducible deletion mESCs and somatic cells

mESC lines with *Upf3a*-inducible deletion were generated and maintained by following a previously published protocol (Li et al, 2015). Briefly, *Upf3a*^f/f Cre-ER^T2+ males were crossed with *Upf3a*^f/f Cre-ER^T2- females. At E3.5, blastocysts were flushed out with 1✕D-PBS into 6-cm sterile Petri dish. Blastocysts were further transferred onto mitomycin C–inactivated ICR MEF feeders. Around 5–7 d, once the inner cell masses/ESCs grew out of blastocysts, they were digested with 0.25% trypsin and further cultured on feeders. The genotyping of the sex of each ESC line was conducted with published primers (Tunster, 2017).

Somatic cells from mouse rib muscle tissues (RDSCs) were generated with finely chopped rib muscle tissues from male *Upf3a*^f/f Cre-ER^T2+ mice at 4 wk of age. The fine-chopped rib muscle tissues were attached to surfaces of 60-mm cell culture dishes and cultured in EF medium (high glucose DMEM, supplemented with 10% FBS, 100 U/ml penicillin, and 100 µg/ml streptomycin). Around 5–7 d, when RDSCs grew out attached tissues, they were digested with 0.05% trypsin and expanded in EF medium.

For the deletion of UPF3A in mESCs and RDSCs, 4-OHT (H6278, 1 µM; Sigma-Aldrich) was used to treat cells for five successive days.

To measure the proliferation of ESCs, $1 \times 10^6$ ESCs were plated in 6-well plates precoated with 0.1% gelatin. The number of ESCs was counted every 3 d.

## Expressions of mouse UPF3A and UPF3B in U2OS cells

cDNAs of mouse *Upf3a* and *Upf3b* were amplified with PrimeSTAR HS Premix (Takara) from an E14.1 ES cell cDNA library. The PCR products were further cloned in a pEGFP-C1-EF1a vector with a seamless cloning kit (D7010S, Beyotime) (Li et al, 2015). Sequences of pEGFP-C1-EF1a-mUpf3a and pEGFP-C1-EF1a-mUpf3b were validated through sequencing services provided by Tsingke. These plasmids were amplified with E.Z.N. A Endo-free plasmid Midi kit (Omega BIO-TEK) and transiently transfected into U2OS cells with Lipofectamine 3000 (Invitrogen) following the company protocols. Protein samples were harvested 48 h after transfection.

## Protein extraction and analysis

Cells or mouse tissues were lysed with RIPA buffer (20-188; Sigma-Aldrich) supplemented with combinations of protease/phosphatase inhibitors (APExBIO). Around 40–60 µg protein was separated with gradient SDS–PAGE gel (4–20%, ACE Biotechnology). The following primary antibodies were used: rabbit anti-UPF3A+ UPF3B (ab269998, Abcam, 1:1,000); mouse anti-$\beta$-actin (A5441; Sigma-Aldrich, 1:10,000); mouse anti-Lamin B1 (sc-374015; Santa Cruz). The secondary antibodies used in these studies were HRP-conjugated goat anti-rabbit IgG or goat anti-mouse IgG (1:2,000; Proteintech).

## qRT–PCR

Cells or mouse tissues were lysed with TRIzol Reagent (Sigma-Aldrich), and total RNAs were purified according to the company protocol. cDNAs were synthesized using HiScript III 1st Strand cDNA Synthesis Kit (R312; Vazyme) according to the company manual. qRT–PCR in triplicate for each sample was performed using 2xTSINGKE Master qPCR Mix (SYBR Green I) (TSINGKE) on the CFX96 Real-Time PCR system (Bio-Rad). The expression of $\beta$-actin was used as the internal control.

Four sets of primers that amplify different parts of the *Upf3a* gene were used to quantify the *Upf3a* mRNA expression in control and *Upf3a* KO ESCs. These primers are:

*Upf3a-E1/E2*: F, CCCTAAGTGAGAGCGGGG;
R, CTCTTCCAGCTGCTCTTTGG;
*Upf3a-E2/E3*: F, GCGCACGATTACTTCGAGGT;
R, TCAAAACGGTCTCTGAACAGC;
*Upf3a-E7/E8*: F, GTAAGAGGAAGGAGGCGGAG; R, TTTCTCTGTGGCCACTTCCT;
*Upf3a-E8/E9*: F, TGGAGACGAGAAGCAGGAAG; R, AGATCTCTTGTCCCTTGGCT.

To quantify the NMD efficiency with qPCR, 23 NMD target genes were selected: *Snhg12*, *Atf4*, *Gas5*, *1810032O08Rik*, and *Ddit3* (Weischenfeldt et al, 2008); *Cdh11*, *Ire1*, *Smad5*, *Smad7*, and *Snord22* were retrieved from Shum et al (Shum et al, 2016); *Auf1* and *Hnrnpl* were from Li et al (Li et al, 2015); *Eif4a2* (PTC isoform) was from Huth et al (Huth et al, 2022). Other gene-specific primers designed and used in this study are listed as bellow:

*Smg5*: F, GGAACTGCTGTGGAGAAAGG; R, AGCGACCAGATGAGTCCTGT;
*Smg6*: F, GAGAACCCAGAGCAGATTCG; R, CAAGCCCATCCATGTAGTCC;
*Smg7*: F, AACCCAAATCGAAGTGAAGTCC; R, ACACCGTACACAGTTCCTGTAA;
*Upf3b*: F, AGGAGAAACGAGTGACCCTGT; R, CCTGTTGCGATCCTGCCTA;
*Eif4a1-PTC*: F, GGGTCGGACGCTCTATAAGT;

R, GTCGGGGCCATTGTCTCT.
*Mettl23-PTC*: F, ACCCAGCTCTTTCGGTTCC; R, AGGAGGGATTAAGGGCATGG;
*Sfrs10-PTC*: F, TTCAGGAAAGGCCCGTAGC; R, TGTCAAATGACGACTTCCGC;
*Snrpb-PTC*: F, AGAAGCCTCTGACCCTCTTCA; R, GACCAGGTTCTCCCCTCGAA;
*Hnrnpa2b1-PTC*: F, GGTGGCTATGGTGGAAGGAG; R, TACAGTCTT-TGTGGCAGCAGA;
*Luc7l-PTC*: F, GGGAGTTGCAGAAAAGCCTC; R, TGACTCTTGCAGACACGGTC;
*Pkm2-PTC*: F, CAGCGTGGAGGCCTCTTATA; R, AAGTGGTAGATGGCAGCCTC.

For semi-quantitative RT–PCR analysis to identify the normal and the PTC$^+$ isoforms of *Pkm2*, *Rps9*, and *Ptbp2*, primer sequences were retrieved from the previous study (Weischenfeldt et al, 2012). Primer sequences for *Eif4a2*, *Luc7l*, *Snrpb*, *Hnrnpa2b1*, *Sfrs10*, *Alkbh3*, *Sf1*, *Nfyb*, *Ccar1*, *Slc38a2*, and *Flot1* were described previously (McIlwain et al, 2010).

## Knockdown of *Upf3a* and *Upf3b* in mESCs

To knockdown *Upf3a* and *Upf3b* in mESCs, control (non-target, NT-siRNA) (50 pmol), mouse *Upf3a* siRNAs (50 pmol) and *Upf3b* (50 pmol) siRNAs were transfected by Lipofectamine RNAiMAX reagent (Invitrogen) into *Upf3a*-proficient and *Upf3a*-deficient mESCs with a previously published reverse transfection method (Li et al, 2015). The mESCs were maintained in the LIF-free differentiation condition. Twelve hours after the siRNA transfection, the medium was changed to the fresh ES medium and the cells were incubated for additional 48 h. RNAi efficiency was investigated by qPCR and Western blotting around 60 h after the siRNA transfection.

The following siRNA sequences are used:
Upf3a-1: 5'-AGAGAAACCCAAAGAAAGA-3';
Upf3a-2: 5'-ACAGGATACCAGTGATAAA-3';
Upf3a-3: 5'-GAGCACAAGGAGTATGACA-3';
Upf3b-1: 5'-CCAAGAGACTGGACAAAGA-3';
Upf3b-2: 5'-GCATGATCCGAGAAAGAGA-3',
Upf3b-3: 5'-GAGTGAGAATACAGAATCA-3'.
All siRNAs are synthesized by RIBOBIO co.

## Statistical analysis

The unpaired *t* test was used in this study. The statistical analysis in this study was performed with GraphPad Prism (Ver 9.00; GraphPad Software).

# Supplementary Information

# Acknowledgements

We thank all members of the Li laboratory for discussion and reading of the article. We thank Dr. Haiyan Yu, Ms. Yuyu Guo, and Ms. Xiaomin Zhao (Core Facilities for Life and Environmental Sciences, State Key laboratory of Microbial Technology, Shandong University) for helping with microscopy. Research projects in laboratory of Tangliang Li are supported by Grant No. LY22C050003 from the Zhejiang Provincial Natural Science Foundation of

China (ZJNSF); Grant No. KF2020005 and KF2021008 from NHC Key Laboratory of Birth Defect for Research and Prevention (Hunan Provincial Maternal and Child Health Care Hospital, Changsha, China); Grant No. 31770871 from National Natural Science Foundation of China, and Qilu Youth Scholar Startup Funding of Shandong University.

## Author Contributions

C Chen: data curation and investigation.
Y Shen: data curation and investigation.
L Li: data curation and investigation.
Y Ren: resources.
Z-Q Wang: resources.
T Li: conceptualization, data curation, supervision, funding acquisition, and writing—original draft.

## Conflict of Interest Statement

The authors declare that they have no conflict of interest.

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
