## [Reviewer comments · Life Science Alliance]

Life Science Alliance

UPF3A is dispensable for nonsense-mediated mRNA decay in mouse pluripotent and somatic cells

Chengyan Chen, Yanmin Shen, Luqian Li, Yaoxin Ren, Zhao-Qi Wang, and Tangliang Li

DOI: <https://doi.org/10.26508/lsa.202201589>

Corresponding author(s): Tangliang Li, Shandong University

Review Timeline:

Submission Date:	2022-07-04
Editorial Decision:	2022-08-02
Revision Received:	2023-02-20
Editorial Decision:	2023-03-14
Revision Received:	2023-03-20
Accepted:	2023-03-20

Scientific Editor: Novella Guidi

Transaction Report:

August 2, 2022

Re: Life Science Alliance manuscript #LSA-2022-01589-T

Dr. Tangliang Li
State Key Laboratory of Microbial Technology, Shandong University
No. 72, Binhai Road
Hangzhou, Zhejiang 311121
China

Dear Dr. Li,

Thank you for submitting your manuscript entitled "Upf3a is dispensable for nonsense-mediated mRNA decay in mouse pluripotent and somatic cells" to Life Science Alliance. The manuscript was assessed by expert reviewers, whose comments are appended to this letter. We invite you to submit a revised manuscript addressing the Reviewer comments.

Thank you for this interesting contribution to Life Science Alliance. We are looking forward to receiving your revised manuscript.

Sincerely,

B. MANUSCRIPT ORGANIZATION AND FORMATTING:

Reviewer #1 (Comments to the Authors (Required)):

Upf3a is dispensable for nonsense-mediated mRNA decay in mouse pluripotent and somatic cells

Chen et al.

The roles of Upf3a and Upf3b in nonsense-mediated mRNA decay (NMD) in mammals remain unclear. In 2016, Shum et al provided evidence for a repressive role for Upf3a in NMD in multiple mouse cell types. More recently, however, two studies published by Wallmeroth et al and Yi et al using whole transcriptomic analyses in human cells strongly support a functional redundancy between Upf3a and Upf3b in NMD and that Upf3a functions similarly to Upf3b in activating NMD. In the current manuscript, Chen et al assess the role of Upf3a in NMD in adult mice tissues by generating an inducible CRE Upf3a knockout mouse (thereby avoiding the embryonic lethal phenotype previously observed in Upf3a knock-outs). Using this resource, the authors demonstrate loss of Upf3a in mouse embryonic stem cells does not result in either the accumulation or enhanced degradation of NMD-targeted transcripts. Additionally, RT-qPCR analysis of a panel of known NMD substrates (and controls) in several somatic cell types derived from the conditional Upf3a KO mouse also indicated that, for the vast majority of target transcripts tested, loss of Upf3a neither enhanced nor repressed NMD in these cells. These data are wholly consistent the recent findings for Upf3a/b function in NMD and add little to our understanding of how this protein paralog functions in NMD. In principle, while the findings should be shared with the community, a more detailed discussion of the Wallmeroth and Yi papers, greater rationale for the experimental design, and better description of how the current data was analyzed would be needed to enhance the interest in these report.

Major Comments:

- The authors provide only a cursory mention of the most recent literature on Upf3a function, and fail to provide a full description of the Wallmeroth et al, 2022 and Yi et al, 2022 papers and how their current results fit in the context of these two studies.
- While many of the methods used here have been previously published, brief descriptions of these protocols should be included. This includes, but is not limited to, generation of the Upf3a conditional knockout mouse and the generation of mESC lines. For this reviewer, the following issues need to be clarified.
 - o Which primers were used to assess the nature of the Upf3a allele (wild-type, floxed, and deletion samples) and why is it that the Upf3a KO allele leads to a larger PCR amplicon than both the wild-type and floxed Upf3a alleles (Figure 1C, Supp Figure 2A). Additionally, how does deletion of Upf3a exon 3 induce Upf3a loss of function (is the transcript now an NMD target).
 - o The authors indicate that all mESC lines were genotyped as female and RDSCs were derived from adult male mice. The use of only one sex in the analysis of each of these cells is unclear and needs to be justified or explained in the text.
 - o The authors analyze the accumulation of 20 NMD-targeted transcripts in multiple cell types; however not all genes were analyzed in each cell type, and the authors do not provide sufficient explanation for selection of their targets in the text.
- The authors conclude that Upf3a KO in mESCs and RDSCs do not lead to phenotypic changes relative to parent/control cells (page 9: Upf3a KO mESCs "show no phenotypic difference"; page 10: "loss of Upf3a does not cause visible cell viability changes"). How was viability and cell phenotype measured and what data support this conclusion (other than similar gross cell morphology by microscopy).

Minor Comments:

- This study does not address whether Upf3a knockout impacts "embryonic stem cell self-renewal or differentiation" (page 4)
- It is unclear why the authors presented a Western blot analysis of GFP-tagged Upf3a and Upf3b expressed in U2OS cells; it appears that the data was never referred to in the text.
- The discussion states that "future transcriptome-wide analysis in Upf3a KO cells and mice would precisely resolve Upf3a's role in NMD"; this is precisely the approach employed by Wallmeroth et al, 2022 and Yi et al, 2022.
- This manuscript would benefit from editing for spelling, grammar, and clarity.

Reviewer #2 (Comments to the Authors (Required)):

The study entitled "Upf3a is dispensable for nonsense-mediated mRNA decay in mouse pluripotent and somatic cells" by Chen, Shen et al. from the Li lab primarily addresses the question how the nonsense-mediated mRNA decay (NMD) factor Upf3a influences NMD activity in different mouse cell types and tissues. NMD is best characterized as a conserved eukaryotic quality control mechanism that identifies and degrades mRNA containing premature translation termination codons. In vertebrates, the two paralogous genes Upf3a and Upf3b are involved in this process and their functional contribution is currently debated in the field. Although initial studies indicated that both Upf3a and Upf3b can act as activators of NMD, a later study using various murine cell types and tissues provided evidence for antagonistic roles for both NMD factors, with Upf3a acting rather as a general NMD repressor (Shum et al. 2016).

In this study, the authors generated conditional Upf3a knockout mice and test the NMD activity in multiple cell types (embryonic stem cells and rib muscle-derived somatic cells) and tissues (liver, spleen and thymus). Two experimental approaches were used to monitor NMD activity: quantitative RT-PCR of previously identified NMD targets and semi-quantitative RT-PCR to distinguish PTC-containing alternatively spliced isoforms from the normal spliced isoform. Overall, the authors found no compelling evidence supporting the proposed NMD-repressing function of Upf3a, thus contradicting the findings of Shum et al. 2016. However, the results are in good agreement with two recent publications from this year (Yi et al. 2022 and Wallmeroth et al. 2022), which both found no/minimal evidence for the antagonistic function of UPF3A in human cell lines.

Although the study by Chen, Shen et al. is rather descriptive and does not provide significant novel mechanistic insights for NMD, it is of high quality and overall important to the field, since it clarifies the NMD-related role of Upf3a in mice. In combination with the recent literature, it will help to paint a more complete picture about the functional role of this NMD factor in mammals.

Main points:

1. Experimental approach to knockout Upf3a

The authors used a similar (or even identical) approach to knockout Upf3a as has been used in the Shum et al. 2016 study, which allows to compare results from both studies more directly. This conditional knockout strategy of floxing exon 3 of Upf3a also circumvents the embryonic lethality of Upf3a full knockouts.

Since all major conclusions in the study are based on this successful knockout approach and no orthogonal methods (other KO strategy, knockdown, ...) were used, it is critical to show that the investigated Upf3a knockout cells indeed did not express any functional Upf3a (full-length or partially truncated) proteins. The authors used an antibody for western blot analyses that is - according to the manufacturer - able to detect both Upf3a and Upf3b. Although the results in Supplementary Figure 1A are convincing, Figure 1D raises a few issues:

- Lane 2 (mESC, + 4-OHT) appears to show residual Upf3a protein expression as faint band in the "long exp." panel.
- Lane 4 (RDSC, +4-OHT) appears weak in intensity (even in the longer exposure), probably due to the lower Upf3b expression in RDSC. This makes it somewhat difficult to estimate if "Upf3a protein was completely absent" or not.
- All western blots are cropped below the Upf3a band, which prohibits judging if truncated Upf3a proteins are expressed as result of exon 3 deletion. Although exon 3 deletion should result in a frameshift, misregulation on the splicing or translation level could lead to production of partially functional proteins (see e.g.: <https://doi.org/10.1038/s41467-019-12028-5>).
- In human cells, UPF3A normally runs higher than UPF3B in western blot analyses. Therefore, the labeling on the right for "hUpf3a" and "hUpf3b" are misleading and should be corrected.

Due to the points raised above, additional experiments/repetitions are suggested (see below). At least one of these approaches (or alternatives) should be tested to substantiate the full Upf3a knockout conclusion.

- Use other commercially available Upf3a-specific antibodies and show with these antibodies to confirm that Upf3a protein is not detectable in the Upf3a knockout cell types and tissues. This approach serves the additional purpose that two different antibodies normally detect different epitopes and therefore complement each other. This approach should take 1-2 weeks depending on availability of other antibodies.
- Repeat the western blots with at least the mouse-derived samples, aiming for a more intense signal for Upf3a. If residual Upf3a signal is detectable, the authors should discuss and interpret this accordingly. This could be done in 1 week if the original samples are still available or multiple weeks if the samples have to be harvested again.
- Show full western blot images at least in the Supplementary Figures to show whether truncated Upf3a proteins are expressed.
- The authors used exogenously expressed mUpf3a and mUpf3b to show the specificity of the used antibody, which is required since the antibody is only described as "Reacts with: Human, Zebrafish" by the manufacturer, thus reactivity with mouse proteins was not tested/validated. Another approach (expected 2-4 week timeframe) to show the specificity could be a series of siRNA-mediated knockdowns (or similar experimental strategies) to downregulate a) Upf3a and/or b) Upf3b in e.g. control mESC or RDSC. Western blot analyses should then confirm the identity of the indicated bands in Figure 1D. At the moment, it is not clear that the upper band in lanes 1-4 in Figure 1D is truly Upf3b.
- To exclude aberrant splicing patterns due to exon 3 deletion, the Upf3a mRNA could be analyzed using the existing cDNA samples (control vs. knockout) with semiquantitative RT-PCR and primers located in the first and last exon. The Ensembl database lists only 1 fully spliced isoform for Upf3a, so the results should be clearly interpretable. In 1-2 weeks this experiment could be performed.
- As last resort - if western blot analyses fail - targeted or whole proteome mass spectrometry could be employed to detect

Upf3a-derived peptides. However, this approach is technically demanding and very time-consuming.

2. Measuring NMD activity

The main conclusion of the study is that loss of Upf3a in various cell types and tissues does not lead to substantially altered NMD activity. More specifically, the authors claim to find no evidence for Upf3a acting as an NMD repressor, but rather observe unchanged or slightly upregulated levels of known NMD targets upon Upf3a knockout.

The presented data based on quantitative and semiquantitative RT-PCR are overall convincing and representative, since a broad selection of previously known NMD targets was chosen. Furthermore, the usage of Smg6 knockout cells as positive control provides an indication of how strong the observable effect of NMD inhibition could be for each individual target.

One major concern is that the loss of Upf3a was analyzed without investigating the impact of the other paralog Upf3b. The authors conclude based on their presented results that Upf3a is "dispensable for nonsense-mediated mRNA decay", which is presumably accurate as long as Upf3b supports NMD activation. The established Upf3a KO cells would serve as an ideal background system to additionally deplete Upf3b (e.g. by siRNA treatment) and examine if NMD activity is severely impaired. This would increase the overall impact of the study overall and help to clarify the interdependence of Upf3a and Upf3b. It would also allow to determine which of the used NMD targets in this study are really Upf3 (Upf3a+Upf3b)-dependent. It is advised that this Upf3b co-depletion should only be performed for selected cell types.

A few suggestions/comments concerning the NMD measurements in general:

- The results of the different cell types and tissues are shown and discussed one after the other. This makes it easier to follow the results, but in the end, it could help the reader to estimate the extend of NMD-related changes if the authors could provide a kind of "top-level" aggregated analysis of their PCR results. Since 20 qPCR targets were measured in multiple cell lines (sometimes with additional controls) it is hard to directly compare those different results (also since the y-axis scales are different). If possible, the authors could show the relative expression for all tested conditions (cell type ~ target), including the Smg6 KO controls, side-by-side in e.g. a heatmap or similar representation.
- Additionally, it could help to provide more qualitative information about the used targets. How many and which of the used target are (significantly) upregulated in the Smg6 KO cells? Which targets are significantly upregulated at least in one Upf3a KO cell line/type? Which are significantly downregulated (seems to apply only to spleen samples)? The authors could implement such information in the overview mentioned above.
- The qPCR data for Upf3a expression could be misleading, since the used reverse primer binds in the deleted exon 3. This means that the qPCR result rather tests for the presence of Upf3a exon 3 than for the general expression of Upf3a. The indicated very low relative expression in the Figures suggests that Upf3a mRNA in general is not expressed anymore, which is probably not the case. It is suggested to either indicate this exon 3 specificity clearly or repeat the analysis with a non-exon 3 specific primer pair (which would complement the existing data).
- The semiquantitative RT-PCRs are a good alternative to estimate NMD activity. However, it is currently only used qualitatively and the reader has to visually inspect each panel. Due to the large number of tested targets, it may not be easy to show the quantification of the PTC-containing bands, but the authors could consider including such information/plots in the supplements.
- The quantification of the gel results in Figure 3 and Supplementary Figure 3 would also help to estimate if PTC-containing isoforms are less abundant in certain Upf3a KO cells compared to controls. This would test for putative Upf3a NMD repressor activity. At the moment the authors simply state that no "detectable change on the PTC isoforms tested" was observed, but this claim should be substantiated with the appropriate quantification.
- For selected AS-NMD targets, the authors could design qPCR primer pairs that separately detect the normal and the PTC-containing isoform, to quantify them more accurately.

Minor points:

- Although the general structure of the manuscript is clear, the text would benefit from another round of carefully checking for typos, confusing sentences and correct grammar (examples are given below). Furthermore, certain statements could be discussed in more detail. To simplify the identification of the text passages, continuous line numbers were inserted into the manuscript text.
 - o The role of Upf3a (and Upf3b) in NMD: On multiple occasions Upf3a is designated by the authors as "dispensable for NMD" (Title, abstract, line 94-95 on page 4, etc.). Considering the results of the authors, it is recommended to extend the discussion about the potential functional redundancy of both Upf3 paralogs. For example, are the rather mild effects on NMD activity upon loss of Upf3a due to Upf3b maintaining NMD?
 - o Examples for typos or confusing sentences:
 - o Line 44 (page 3): confusing - "... after DNAs are transcribed to form RNAs with RNA polymerase"
 - o Line 147 (page 6) and line 156 (page 7): typo - please correct "lysed" to "lysed"
 - o Line 298 (page 12): typo - please correct "Wallmroth" to "Wallmeroth"
 - Supplementary Figure 2B: the kidney samples do not show any Upf3a or Upf3b expression except for faint bands? Can the authors comment on this or otherwise consider to remove/repeat this western blot?

(This report is from Volker Böhm)

Reviewer #3 (Comments to the Authors (Required)):

In manuscript (MS), Chen et al described that Upf3a is dispensable for NMD in mouse pluripotent and somatic cells. Through generating a conditional Upf3a knockout (KO) mouse mutant, the authors used Upf3a KO stem cell line, somatic cells and liver, spleen and thymus to investigate the functions of Upf3a in NMD pathway. They found that Upf3a was not required for NMD in ESCs, while NMD was impaired in Smg6 KO ESCs. They also showed that obvious differences of AS-NMD targets were observed in Smg6 KO ESCs, but not in Upf3a KO ESCs. Only few NMD targets were upregulated in Upf3a KO liver, thymus and spleen, which suggested that Upf3a is dispensable for NMD in mouse pluripotent and somatic cells.

The conclusion of this MS is different from the observations of Shum et al 2016, which provided lots of evidence to show that Upf3a inhibits degradation of NMD targets. Most of conclusions are supported by the data in this MS. While potentially interesting, there are also major concerns with this report.

Major concerns

1. In this MS, the conclusion was based on only known NMD targets. It is necessary for the authors to perform transcriptome analysis to illustrate Upf3a functions at genome-wide level.
2. UPF3A and UPF3B are two paralogues of yeast Upf3. UPF3B is a widely accepted NMD factor negatively regulating mRNA stability. It is better for the authors to use Upf3b as a functional positive control.
3. Only NMD targets of Atf4 and Ddit3 were downregulated, but some of other NMD targets were upregulated in Upf3a KO spleen tissue from adult mouse. So the conclusion "Upf3a deficiency weakly inhibits NMD in multiple tissues from adult mice" is not appropriate.

Minor issues

1. In figure 1C, the authors used PCR to identify KO line. For better understanding, the authors might provide primers information in figure 1A.
2. In McIlwain et al paper, the level of Eif4a2 normal transcript was much higher than PTC-containing isoform. However, In figure 3A, the Eif4a2 PTC-containing isoform is higher than the normal transcript. Please give the explanation for the difference.

Overview of the revision

We thank all three reviewers for their constructive comments. Following these suggestions, we conducted new experiments and made new figures in the revised manuscript. please note: We included Yaoxin Ren as one of the co-authors since she provided valuable resources (ESC lines) during the revision.

The main changes of the manuscripts are:

Newly included Figures	Summary	Specific answer to reviewer
New Fig 1	Localizations of genotyping primers	Reviewer # 3
New Supp Fig 1	Characterization of Upf3a+Upf3b antibody	Reviewer # 2
New Supp Fig 2	New primer sets to quantify the Upf3- Δ E3 expression in ESCs and RDSCs	Reviewers # 1, #2
New Supp Fig3	Proliferation of ESCs	Reviewer # 1
New Supp Fig4	Primer's annotation	Reviewer # 1
New Supp Fig6	New qPCR primers testing; NMD quantifications in ESCs, RDSCs and tissues	Reviewers # 2, # 3
New Supp Fig7	Co-depletion of UPF3A/UPF3B inhibits NMD	Reviewer # 2

Furthermore, we have included all original pictures for DNA and protein gels for evaluation. We consulted English-speaking colleagues to correct all typos. We hope the revision will solve all the concerns raised.

The detailed "point to point" responses are listed below.

Response to Reviewer #1:

Upf3a is dispensable for nonsense-mediated mRNA decay in mouse pluripotent and somatic cells

The roles of Upf3a and Upf3b in nonsense-mediated mRNA decay (NMD) in mammals remain unclear. In 2016, Shum et al provided evidence for a repressive role for Upf3a in NMD in multiple mouse cell types. More recently, however, two studies published by Wallmeroth et al and Yi et al using whole transcriptomic analyses in human cells strongly support a functional redundancy between Upf3a and Upf3b in NMD and that Upf3a functions similarly to Upf3b in activating NMD. In the current manuscript, Chen et al assess the role of Upf3a in NMD in adult mice tissues by generating an inducible CRE Upf3a knockout mouse (thereby avoiding the embryonic lethal phenotype previously observed in Upf3a knock-outs). Using this resource, the authors demonstrate loss of Upf3a in mouse embryonic stem cells does not result in either the accumulation or enhanced degradation of NMD-targeted transcripts. Additionally, RT-qPCR analysis of a panel of known NMD substrates (and controls) in several somatic cell types derived from the conditional Upf3a KO mouse also indicated that, for the vast majority of target transcripts tested, loss of Upf3a neither enhanced nor repressed NMD in these cells. These data are wholly consistent the recent findings for Upf3a/b function in NMD and add little to our understanding of how this protein paralog functions in NMD. In principle, while the findings should be shared with the community, a more detailed discussion of the Wallmeroth and Yi papers, greater rationale for the experimental design, and better description of how the current data was analyzed would be needed to enhance the interest in these report.

Response: We thank to the reviewer that he or she is generally positive on our findings. He or she agreed that our finding, together with Wallmeroth and Yi's papers, all reach a conclusion that UPF3A does not repress NMD. He or she strongly suggested that we should strength the discussions on Wallmeroth and Yi's findings with our data. In the revised manuscript, we revised the introduction, materials and methods, and discussion extensively on the UPF3A functions in NMD (marked in red).

Major Comments:

- The authors provide only a cursory mention of the most recent literature on Upf3a function, and fail to provide a full description of the Wallmeroth et al, 2022 and Yi et al, 2022 papers and how their current results fit in the context of these two studies.

Response: Thanks for this suggestion from this reviewer. We have revised the introduction, and discussion (Page 4-5, Line 102-127, in red) on Wallmeroth and Yi's findings.

- While many of the methods used here have been previously published, brief descriptions of these protocols should be included. This includes, but is not limited to, generation of the Upf3a conditional knockout mouse and the generation of mESC lines. For this reviewer, the following issues need to be clarified.

Response: We thank for this suggestion from this reviewer. In revised manuscript, we have included the details on generation ESCs from mouse E3.5 blastocysts, and RDSCs from adult mice (Page 8, line 200-220). Furthermore, we added the details on siRNA experiments (Page 11-12, line 295-312).

o Which primers were used to assess the nature of the *Upf3a* allele (wild-type, floxed, and deletion samples) and why is it that the *Upf3a* KO allele leads to a larger PCR amplicon than both the wild-type and floxed *Upf3a* alleles (Figure 1C, Supp Figure 2A). Additionally, how does deletion of *Upf3a* exon 3 induce *Upf3a* loss of function (is the transcript now an NMD target).

Response: Thanks to the reviewer for this comment. The primers used for routine genotyping of wild type, floxed and deleted allele are: F1, AAAGAACAGTGTGCAATTACTCGG (before the left LoxP at intron 2); R1, TTCACAGGTAGGAACGATTCCATT (after the left loxP at intron 2); M1, AGCTCTTACTCTTGAGCCCAC (after the right loxP at intron 3). The locations of these primers now are marked in new Fig 1A. In principle, with normal genotyping PCR, F1-R1 primers around left loxP site are ideal to amplify wild type (133bps) and floxed allele (200bps); when *Upf3a* exon 3 is deleted, F1-M1 primers will amplify a PCR band around 412bps, indicating *Upf3a* exon 3 deletion. The M1 primer is located the right side of the right LoxP site, thus the amplicon size is dependent on primer design strategy.

Furthermore, the reviewer asked whether *Upf3a* exon 3 removal will cause function loss of mUPF3A. In our study, we have followed the original *Upf3a* cKO design strategy as described in Shum et al. 2006 (Cell). In the Cell paper, the authors claimed that *Upf3a* exon 3 deletion will break the UPF2 and EJC interacting domains of UPF3A, which will result in UPF3A function loss.

Here, we checked the consequences of *Upf3a* exon 3 deletion. The deletion of *Upf3a* exon3 (*Upf3a*- Δ E3) generates a frameshift of *Upf3a* mRNA, and consequently produces a PTC on *Upf3a* mRNA exon 4. In principle, mRNAs of *Upf3a*- Δ E3 is a NMD target. We could confirm that *Up3a* KO ESCs, RDSCs, and tissues have complete loss of *Upf3a* exon3 if we used qPCR primers located on *Upf3a* exons 2-3. Furthermore, we could detect significant reductions (around 80%) of *Upf3a*- Δ E3 mRNAs if we used qPCR primers on exons 1-2, exons 7-8, and exons 8-9 (see New Supp Figure 2) in ESCs and RDSCs. However, due to UPF3A is largely dispensable for NMD in mice (our major finding in this manuscript), we think mouse *Upf3a*- Δ E3 mRNA transcript is degraded. These findings are documented in Line 378-384 at page 14.

o The authors indicate that all mESC lines were genotyped as female and RDSCs were derived from adult male mice. The use of only one sex in the analysis of each of these cells is unclear and needs to be justified or explained in the text.

Response: We thank the reviewer for this question. In our study, we, in total, generated 6 *Upf3a* inducible ESCs lines (5 females, 1 male). However, due to the "differentiation-like"

morphology of one female and one male ESC lines, we thus used 4 independent ESC lines with good morphology to conduct the study. We introduced a sentence “we isolated E3.5 blastocysts and established several *Upf3a* inducible deletion ESC lines. Four female ESC lines showing the typical ESC morphology were selected for further analysis” at Line 376-378 (Page 14).

Since during times of our study, another research project of lab is male fertility, thus, the majority of mice we kept are male *Upf3a^{fl/fl}Cre-ERT2⁺*. Thus, we only established male RDSC lines to investigate UPF3A’s role in NMD.

o The authors analyze the accumulation of 20 NMD-targeted transcripts in multiple cell types; however not all genes were analyzed in each cell type, and the authors do not provide sufficient explanation for selection of their targets in the text.

Response: We thank the reviewer for these questions. In original Figure 2, we quantified the expression of 19, 19, 17, 16, and 19 NMD targets in ESC, RDSC, liver, thymus and spleen samples in control and *Upf3a* background, respectively. We did not show qPCR data on *Cdh11* and *Smad5* in liver, *Smad7*, *Mettl23*, and *Snhg12* in thymus due to the fact that we found the melting curves of these genes have two peaks. We think “two peaks in melting curve” would cause inaccurate quantification on these target genes expression. For the selection criteria (NMD features, etc.) of these analyzed genes, we have made a table (Supp Figure 4). We add a sentence to refer this figure in revised manuscript (Page 14-15, Line 393-394).

• The authors conclude that *Upf3a* KO in mESCs and RDSCs do not lead to phenotypic changes relative to parent/control cells (page 9: *Upf3a* KO mESCs “show no phenotypic difference”; page 10: “loss of *Upf3a* does not cause visible cell viability changes”). How was viability and cell phenotype measured and what data support this conclusion (other than similar gross cell morphology by microscopy).

Response: This review point is valid. We have compared the proliferation of *Upf3a* KO ESCs with their individual parental ESC lines and found they have no difference (Supp Figure 3).

Minor Comments:

• This study does not address whether *Upf3a* knockout impacts “embryonic stem cell self-renewal or differentiation” (page 4)

Response: This is a brilliant comment from the reviewer. Current data (morphology and proliferation) do not reveal UPF3A’s role in ESC differentiation. Since Wallmeroth and Yi’s papers and our findings all showed that UPF3A do not repress NMD. Since *Upf3a* KO causes early embryonic lethality (Shum et al. Cell, 2016), the next key question in UPF3A biology is whether *Upf3a* KO disturbs ESC self-renewal and differentiation. Currently, UPF3A in ESC self-renewal and differentiation is already in our research pipeline.

- It is unclear why the authors presented a Western blot analysis of GFP-tagged Upf3a and Upf3b expressed in U2OS cells; it appears that the data was never referred to in the text.

Response: We thank the reviewer for this suggestion. The expression of GFP-tagged mUpf3a and mUpf3b in U2OS cells was another evidence to show that the antibody used in our study could detect mUpf3a and mUpf3b. In the revised manuscript, we included a new result section on Upf3a+Upf3b antibody characterization (Supp Figure 1) and explained the reason to express GFP-mUpf3a and GFP-mUpf3b in U2OS (Page 13-14, line 339-368). Furthermore, in the revised figure legend for new Fig 1, a sentence "Protein lysates from U2OS cells expressing GFP-mUpf3a and GFP-mUpf3b are used to validate antibody" is included.

- The discussion states that "future transcriptome-wide analysis in Upf3a KO cells and mice would precisely resolve Upf3a's role in NMD"; this is precisely the approach employed by Wallmeroth et al, 2022 and Yi et al, 2022.

Response: We are sorry for the misunderstanding here. In current manuscript we only used qPCR/semi-quantitative RT-PCR to evaluate the *Upf3a* function in NMD, our next step will concentrate on specific types of mouse cells and tissues, including ESCs, NSCs, and testes, and use transcriptome approaches to define the UPF3A's roles in NMD. In the revised manuscript, we have replaced this sentence with "Future transcriptome-wide analysis on *Upf3a* deficient murine cells or tissues would precisely resolve UPF3A's role in NMD" (Page 20, line 549-551).

- This manuscript would benefit from editing for spelling, grammar, and clarity.

Response: Thanks for the suggestion, we have consulted the help for English-speaking colleagues and revised our manuscript.

Response to Reviewer #2:

(This report is from Volker Böhm)

The study entitled "Upf3a is dispensable for nonsense-mediated mRNA decay in mouse pluripotent and somatic cells" by Chen, Shen et al. from the Li lab primarily addresses the question how the nonsense-mediated mRNA decay (NMD) factor Upf3a influences NMD activity in different mouse cell types and tissues. NMD is best characterized as a conserved eukaryotic quality control mechanism that identifies and degrades mRNA containing premature translation termination codons. In vertebrates, the two paralogous genes Upf3a and Upf3b are involved in this process and their functional contribution is currently debated in the field. Although initial studies indicated that both Upf3a and Upf3b can act as activators of NMD, a later study using various murine cell types and tissues provided evidence for antagonistic roles for both NMD factors, with Upf3a acting rather as a general NMD repressor (Shum et al. 2016).

In this study, the authors generated conditional Upf3a knockout mice and test the NMD activity in multiple cell types (embryonic stem cells and rib muscle-derived somatic cells) and tissues (liver, spleen and thymus). Two experimental approaches were used to monitor NMD activity: quantitative RT-PCR of previously identified NMD targets and semi-quantitative RT-PCR to distinguish PTC-containing alternatively spliced isoforms from the normal spliced isoform. Overall, the authors found no compelling evidence supporting the proposed NMD-repressing function of Upf3a, thus contradicting the findings of Shum et al. 2016. However, the results are in good agreement with two recent publications from this year (Yi et al. 2022 and Wallmeroth et al. 2022), which both found no/minimal evidence for the antagonistic function of UPF3A in human cell lines.

Although the study by Chen, Shen et al. is rather descriptive and does not provide significant novel mechanistic insights for NMD, it is of high quality and overall important to the field, since it clarifies the NMD-related role of Upf3a in mice. In combination with the recent literature, it will help to paint a more complete picture about the functional role of this NMD factor in mammals.

Response: We thank to the reviewer (Dr. Volker Böhm) for his positive comments on our findings. These thoughtful suggestions, which provide detailed experimental designs to improve our manuscript, are highly appreciated by all authors. We have revised the manuscript based on his suggestions. The details are listed below.

Main points:

1. Experimental approach to knockout Upf3a

The authors used a similar (or even identical) approach to knockout Upf3a as has been used in the Shum et al. 2016 study, which allows to compare results from both studies more directly. This conditional knockout strategy of floxing exon 3 of Upf3a also circumvents the embryonic lethality of Upf3a full knockouts.

Since all major conclusions in the study are based on this successful knockout approach and no orthogonal methods (other KO strategy, knockdown, ...) were used, it is critical to show that the investigated Upf3a knockout cells indeed did not express any functional Upf3a (full-length or partially truncated) proteins. The authors used an antibody for western blot analyses that is - according to the manufacturer - able to detect both Upf3a and Upf3b. Although the results in Supplementary Figure 1A are convincing, Figure 1D raises a few issues:

- Lane 2 (mESC, + 4-OHT) appears to show residual Upf3a protein expression as faint band in the "long exp." panel.

Response: We thank the reviewer raising this point. As mentioned in the text, we used 4-OHT to treat *Upf3a^{fl/fl}* Cre-ERT2⁺ ESCs for 5 days, which will induce Cre-ER fusion protein translocating from cytoplasm to nucleus to delete *Upf3a* exon 3. Due to the fact that Cre recombinase will not be 100% effective, very little amount ESCs still have intact Upf3a flox allele. This is the reason that if we exposed the blot for longer time, we will see a very faint UPF3A band. We could isolate single *Upf3a* KO ESC clone and do further analysis. However, to maintain a stringent control-KO pair, we did not apply this approach.

- Lane 4 (RDSC, +4-OHT) appears weak in intensity (even in the longer exposure), probably due to the lower Upf3b expression in RDSC. This makes it somewhat difficult to estimate if "Upf3a protein was completely absent" or not.

Response: This is a good point, which is related to distinct expression pattern of UPF3A and UPF3B among different cells and tissues. In Fig 1D, lines 1-2 are samples from ESCs, and Lines 3-4 from somatic cells (RDSCs), and they do express different amount of UPF3 paralogs. Our data find that ESCs express more UPF3A and UPF3B than RDSCs (if we used the actin to normalize the protein amount loaded). Please also note: the relative intensity of UPF3B vs. UPF3A is different between ESCs and RDSCs. In lane 3 (RDSCs without 4-OHT treatment, UPF3B protein level is also low. Thus, in male RDSCs, cells express less UPF3B than UPF3A.

- All western blots are cropped below the Upf3a band, which prohibits judging if truncated Upf3a proteins are expressed as result of exon 3 deletion. Although exon 3 deletion should result in a frameshift, misregulation on the splicing or translation level could lead to production of partially functional proteins (see e.g.: <https://doi.org/10.1038/s41467-019-12028-5>).

Response: Thanks to the reviewer for reminding that CRISPR-Cas9 strategy could generate partial functional proteins caused by frameshift, splicing mis regulation, etc.. In Tuladhar et al. (Nature Communications, 2019), to generate a KO allele, the strategy is to use CRISPR-Cas9 strategy to make mutations (indels, etc.) in exons of a gene. However, in our case, we specifically modify the intron sequences of *mUpf3a* gene. Furthermore, in our study, short homology oligos designed based on *Upf3a* intron sequences are used to

guide the correct integration of LoxP sites flanking *mUpf3a* exon3, which will make the gene targeting more precisely. Since strategies between Tuladhar et al. and ours are different, it is difficult to compare the outcome. Furthermore, we include the full blots of Fig 1D and Supp Figure 3 to show that UPF3A protein is completely deleted. The original uncropped images of WB are attached during the revision submission.

- In human cells, UPF3A normally runs higher than UPF3B in western blot analyses. Therefore, the labeling on the right for "hUpf3a" and "hUpf3b" are misleading and should be corrected.

Response: The comment of molecular weights of hUPF3A and hUPF3B is valid. In Wallmeroth's EMBO paper, hUPF3A band is closer to 75KD marker than hUPF3B, and Yi's EMBO paper showed hUPF3A protein runs slower than hUPF3B in WB gel. These findings all indicate that hUPF3A has bigger molecular weight than hUPF3B. Since we only use the human U2OS cells to prepare protein lysate with GFP-mUpf3a and GFP-mUpf3b for antibody validation, we removed the marks of "hUpf3a" and "hUpf3b" in revised Fig 1D.

Due to the points raised above, additional experiments/repetitions are suggested (see below). At least one of these approaches (or alternatives) should be tested to substantiate the full *Upf3a* knockout conclusion.

- Use other commercially available *Upf3a*-specific antibodies and show with these antibodies to confirm that *Upf3a* protein is not detectable in the *Upf3a* knockout cell types and tissues. This approach serves the additional purpose that two different antibodies normally detect different epitopes and therefore complement each other. This approach should take 1-2 weeks depending on availability of other antibodies.
- Repeat the western blots with at least the mouse-derived samples, aiming for a more intense signal for *Upf3a*. If residual *Upf3a* signal is detectable, the authors should discuss and interpret this accordingly. This could be done in 1 week if the original samples are still available or multiple weeks if the samples have to be harvested again.
- Show full western blot images at least in the Supplementary Figures to show whether truncated *Upf3a* proteins are expressed.
- The authors used exogenously expressed mUpf3a and mUpf3b to show the specificity of the used antibody, which is required since the antibody is only described as "Reacts with: Human, Zebrafish" by the manufacturer, thus reactivity with mouse proteins was not tested/validated. Another approach (expected 2-4 week timeframe) to show the specificity could be a series of siRNA-mediated knockdowns (or similar experimental strategies) to downregulate a) *Upf3a* and/or b) *Upf3b* in e.g. control mESC or RDSC. Western blot analyses should then confirm the identity of the indicated bands in Figure 1D. At the moment, it is not clear that the upper band in lanes 1-4 in Figure 1D is truly *Upf3b*.
- To exclude aberrant splicing patterns due to exon 3 deletion, the *Upf3a* mRNA could be analyzed using the existing cDNA samples (control vs. knockout) with semiquantitative RT-PCR and primers located in the first and last exon. The Ensembl database lists only 1 fully spliced isoform for *Upf3a*, so the results should be clearly interpretable. In 1-2 weeks this

experiment could be performed.

- As last resort - if western blot analyses fail - targeted or whole proteome mass spectrometry could be employed to detect Upf3a-derived peptides. However, this approach is technically demanding and very time-consuming.

Response: These are excellent suggestions. We did the following two strategies to show that new Abcam antibody works with mouse samples.

- 1) We included the full blot (related to new Fig 1D, and New Supp Fig 1B) to show that Upf3a+Upf3b antibody could recognize GFP-mUpf3a and GFP-mUpf3b protein expressed in U2OS cells. And, this antibody only detects 2 bands, presumably UPF3A and UPF3B in mouse ESC and somatic cells (RDSC). The lower band in ESCs and RDSCs samples is UPF3A, which is supported by the *Upf3a* KO samples.
- 2) We designed new siRNAs against mouse *Upf3a* and *Upf3b* to validate the antibody in ESCs. siRNA-mUpf3b treatment efficiently depletes mUPF3B (upper bands) and causes a dramatic increase of lower bands (mUPF3A), which is consistent with previous findings that UPF3B knockdown or knockout upregulates UPF3A protein levels (Shum et al. Cell, 2006, Wallmeroth et al. and Yi et al. EMBO, 2022, etc.). Thus, we are very confident that the new Upf3a + Upf3b antibody from Abcam could detect mUPF3A and mUPF3B in a single blot and will be of great value to solve the discrepancy on functions of UPF3 paralogs in mammals. Of note, we designed 3 different siRNAs against *mUpf3a*, the knockdown efficiencies are not ideal. However, we still could see these siRNA-mUpf3as reduce the intensity of lower bands, which further confirms the reactivity of Abcam Upf3a+Upf3b antibody. To summarize these data, we included a new section on antibody characterization (please see the new Supp Fig 1; Page 13-14, Line 339-368).
- 3) It is interesting to note, in Supp Fig 1C, in control sample, a faint protein band appears just below protein marker of 52KD, which is missing in siRNA-Upf3a samples and enhanced in siRNA-Upf3b samples. The band may represent a new and uncharacterized isoform of mouse UPF3A. However, since this band is not prominent, we did not discuss it in the text.

2. Measuring NMD activity

The main conclusion of the study is that loss of Upf3a in various cell types and tissues does not lead to substantially altered NMD activity. More specifically, the authors claim to find no evidence for Upf3a acting as an NMD repressor, but rather observe unchanged or slightly upregulated levels of known NMD targets upon Upf3a knockout.

The presented data based on quantitative and semiquantitative RT-PCR are overall convincing and representative, since a broad selection of previously known NMD targets was chosen. Furthermore, the usage of Smg6 knockout cells as positive control provides an indication of how strong the observable effect of NMD inhibition could be for each individual target.

One major concern is that the loss of Upf3a was analyzed without investigating the impact of the other paralog Upf3b. The authors conclude based on their presented results that Upf3a is "dispensable for nonsense-mediated mRNA decay", which is presumably accurate as long as Upf3b supports NMD activation. The established Upf3a KO cells would serve as an ideal background system to additionally deplete Upf3b (e.g. by siRNA treatment) and examine if NMD activity is severely impaired. This would increase the overall impact of the study overall and help to clarify the interdependence of Upf3a and Upf3b. It would also allow to determine which of the used NMD targets in this study are really Upf3 (Upf3a+Upf3b)-dependent. It is advised that this Upf3b co-depletion should only be performed for selected cell types.

Response: We thank the reviewer for the positive comments on methods we used to quantify the NMD efficiency in *Upf3a* KO ESCs, RDSCs and tissues. And these suggestions to test the hypothesis that UPF3B supports NMD activation when UPF3A loss are excellent. Following his suggestion, we measured the NMD efficiency of two of our *Upf3a* inducible knockout ESC lines with *Upf3a/Upf3b* co-depletion. We used semi-quantitative RT-PCR and found that expressions of PTC⁺ isoforms of *Hnmpa2b1*, *Luc7l*, *Pkm2*, *Ptbp2*, *Flot1*, and *Sf1*, are, in general, increased after we used 3 independent siRNA-*Upf3b*s to deplete UPF3B proteins in UPF3A deficient backgrounds (New Supp Fig 7). These data reinforced the conclusion that UPF3B and UPF3A co-depletion could significantly inhibit NMD, as compared with UPF3A or UPF3B single KO or KD. Furthermore, in mice, UPF3A and UPF3B could compensate for each other in NMD. Based on these findings, we included a new "result" section (Page 15-16, line 416-425) and discussed these data in line 538-541 at Page 20.

A few suggestions/comments concerning the NMD measurements in general:

- The results of the different cell types and tissues are shown and discussed one after the other. This makes it easier to follow the results, but in the end, it could help the reader to estimate the extend of NMD-related changes if the authors could provide a kind of "top-level" aggregated analysis of their PCR results. Since 20 qPCR targets were measured in multiple cell lines (sometimes with additional controls) it is hard to directly compare those different results (also since the y-axis scales are different). If possible, the authors could show the relative expression for all tested conditions (cell type ~ target), including the Smg6 KO controls, side-by-side in e.g. a heatmap or similar representation.

the major conclusion is that UPF3A deficiency

- Additionally, it could help to provide more qualitative information about the used targets. How many and which of the used target are (significantly) upregulated in the Smg6 KO cells? Which targets are significantly upregulated at least in one *Upf3a* KO cell line/type? Which are significantly downregulated (seems to apply only to spleen samples)? The authors could implement such information in the overview mentioned above.

Response: We thank to the reviewer on his suggestion to visualize the NMD status in cells and tissues. However, we did not find significant variances (even in fold-change) of gene expressions in cells and tissues. Thus, we only draw a line (at value “1” on Y axis) to make the border mark to compare the relative expressions of NMD targets in control and *Upf3a* KO cells and tissues.

- The qPCR data for *Upf3a* expression could be misleading, since the used reverse primer binds in the deleted exon 3. This means that the qPCR result rather tests for the presence of *Upf3a* exon 3 than for the general expression of *Upf3a*. The indicated very low relative expression in the Figures suggests that *Upf3a* mRNA in general is not expressed anymore, which is probably not the case. It is suggested to either indicate this exon 3 specificity clearly or repeat the analysis with a non-exon 3 specific primer pair (which would complement the existing data).

Response: This point raised by the reviewer is valid. In the original manuscript, we used primers (Forward primer localized on Exon 2, reverse primer localized on Exon 3) to evaluate the *Upf3a* expression and knockout efficiency. These data (presented in Fig 1 and 2) showed the complete deletion of *Upf3a* exon 3. Following the reviewer’s suggestion, we designed another 3 pairs of qPCR primers (spanning exons 1- 2; exons 7–8 and exons 8 -9; primer sequences now are listed in new materials and methods part) and measured the *Upf3a* expressions in ESCs and RDSCs. We found all the qPCR pairs showed that *Upf3a* gene transcripts are significantly reduced (around 80% reduction) (new Supp Fig 2). The deletion of *Upf3a* exon3 (*Upf3a*- Δ E3) generates frameshift of the *Upf3a* mRNA, and consequently produces a PTC on *Upf3a* exon 4 . In principle, mRNAs of *Upf3a*- Δ E3 is a NMD target. The dramatic reduction of *Upf3a*- Δ E3 mRNAs in *Upf3a* KO ESCs and RDSCs further supports our conclusion that UPF3A is dispensable for NMD in mouse when UPF3B is present. We documented this finding in Line 378-384 at Page 14.

- The semiquantitative RT-PCRs are a good alternative to estimate NMD activity. However, it is currently only used qualitatively and the reader has to visually inspect each panel. Due to the large number of tested targets, it may not be easy to show the quantification of the PTC-containing bands, but the authors could consider including such information/plots in the supplements.
- The quantification of the gel results in Figure 3 and Supplementary Figure 3 would also help to estimate if PTC-containing isoforms are less abundant in certain *Upf3a* KO cells compared to controls. This would test for putative *Upf3a* NMD repressor activity. At the moment the authors simply state that no "detectable change on the PTC isoforms tested" was observed, but this claim should be substantiated with the appropriate quantification.
- For selected AS-NMD targets, the authors could design qPCR primer pairs that separately detect the normal and the PTC-containing isoform, to quantify them more accurately.

Response: In these comments, the reviewer suggested to have a quantitative evaluation on semi-quantitative AS-NMD results. In original manuscript, we did not do the quantification. The reasons are: 1) there are almost no appearance of the PTC⁺ isoforms in *Upf3a* KO ESCs, RDSCs and tissue samples; 2) in most cases, it is difficult to capture an unsaturated figure since normal isoforms are always the major bands amplified with PCR. Following the suggestion from reviewer, we designed PTC specific qPCR primers for *Srsf10*, *Snrpb*, *Hnrnpa2b1*, *Luc71* and *Pkm2*. The qPCR results now are shown in Supp Fig 6. We tested these primers with the *Smg6* KO ESC and fibroblast, and found *Smg6* KO increased the expressions of these PTC⁺ isoforms (Supp Fig 6 A). Furthermore, we found in *Upf3a* KO ESC and RDSCs, the mRNA level of these PTC⁺ isoforms are not significantly changed (Supp Fig 6B). In tissues of thymus, spleen, liver and kidney, expressions of these PTC⁺ isoforms are similar (Supp Fig 6C).

Please note: in original manuscript, qPCR primers for *Auf1* and *Hnrnp1* are designed to detect the PTC⁺ isoforms. Thus, for NMD targets with the PTC feature, we totally designed seven sets of gene specific primers, and they all showed that UPF3A deletion have no/minimal effect on NMD in murine cells and tissues.

Minor points:

- Although the general structure of the manuscript is clear, the text would benefit from another round of carefully checking for typos, confusing sentences and correct grammar (examples are given below). Furthermore, certain statements could be discussed in more detail. To simplify the identification of the text passages, continuous line numbers were inserted into the manuscript text.

Response: We thank the reviewer for these suggestions. We have gone through the text and corrected all typos and add new discussions (shown in red).

- o The role of *Upf3a* (and *Upf3b*) in NMD: On multiple occasions *Upf3a* is designated by the authors as "dispensable for NMD" (Title, abstract, line 94-95 on page 4, etc.). Considering the results of the authors, it is recommended to extend the discussion about the potential functional redundancy of both *Upf3* paralogs. For example, are the rather mild effects on NMD activity upon loss of *Upf3a* due to *Upf3b* maintaining NMD?

Response: We thank the reviewer for these great suggestions. In discussion, we have included the discussion point on compensation of UPF3 paralogs in NMD (shown in red, page 20, line 538-541).

- o Examples for typos or confusing sentences:

- o Line 44 (page3): confusing - "... after DNAs are transcribed to form RNAs with RNA polymerase"

Response: we have corrected the sentences "Nonsense-mediated mRNA decay (NMD) is a highly conserved gene expression regulation mechanism after DNAs are transcribed to form RNAs with RNA polymerases. Nonsense-mediated mRNA decay (NMD) surveillances

transcriptome quality by eliminating mRNAs..." to "Nonsense-mediated mRNA decay (NMD) is a highly conserved gene expression regulation mechanism. NMD surveillances transcriptome quality by eliminating mRNAs..." (Page 3, line 60-63).

o Line 147 (page 6) and line 156 (page 7): typo - please correct "lysed" to "lysed"

Response: We have corrected these two typos.

o Line 298 (page 12): typo - please correct "Wallmroth" to "Wallmeroth"

Response: We are sorry for the typo. We have corrected it in the revised text.

• Supplementary Figure 2B: the kidney samples do not show any Upf3a or Upf3b expression except for faint bands? Can the authors comment on this or otherwise consider to remove/repeat this western blot?

Response: We thank the reviewer for carefully reading of our manuscript. Concerning the expressions of UPF3A and UPF3B among tissues, they do show heterogeneities on protein levels among tissues. In kidney tissues, expressions of UPF3A and UPF3B are very low compared with the thymus and spleen. We have deleted kidney blot in revised Supp Fig 8B.

Response to Reviewer #3:

In manuscript (MS), Chen et al described that Upf3a is dispensable for NMD in mouse pluripotent and somatic cells. Through generating a conditional Upf3a knockout (KO) mouse mutant, the authors used Upf3a KO stem cell line, somatic cells and liver, spleen and thymus to investigate the functions of Upf3a in NMD pathway. They found that Upf3a was not required for NMD in ESCs, while NMD was impaired in Smg6 KO ESCs. They also showed that obvious differences of AS-NMD targets were observed in Smg6 KO ESCs, but not in Upf3a KO ESCs. Only few NMD targets were upregulated in Upf3a KO liver, thymus and spleen, which suggested that Upf3a is dispensable for NMD in mouse pluripotent and somatic cells.

The conclusion of this MS is different from the observations of Shum et al 2016, which provided lots of evidence to show that Upf3a inhibits degradation of NMD targets. Most of conclusions are supported by the data in this MS. While potentially interesting, there are also major concerns with this report.

Response: We thank the reviewer for his or her careful evaluation on our manuscript. He or she generally holds faith in our data and finding that UPF3A is dispensable for NMD in murine cells and tissues. This result is different or contradictory to Shum et al. (Cell 2016). In Shum's paper, they found UPF3A in human and mouse, in general, acts as a NMD repressor. However, recently two back-to-back papers from Gehring's and Singh's groups all showed that UPF3A is not a NMD repressor and depletion of UPF3A in human cells does not affect NMD (2022, EMBO J). Our findings here support Gehring's and Singh's findings, and extend the UPF3A's roles in murine cells.

Major concerns

1. In this MS, the conclusion was based on only known NMD targets. It is necessary for the authors to perform transcriptome analysis to illustrate Upf3a functions at genome-wide level.

Response: We thank the reviewer for this comment. Several approaches are used to define NMD in cells and tissues: 1) measuring the NMD efficiency based on expression of known targets (for example, Kurosaki et al. 2021, Nature Cell Biology); 2) using NMD reporter assay (for example, Cho et al. 2022, Mol Cell) 3) transcriptome analysis (RNA-seq or even RIP-Seq) (for examples, Wallmeroth et al. 2022, Yi et al. 2022, EMBO J). And measuring the NMD target expression is a most widely-used approach in defining the NMD efficiency. In our analysis, we selected 7 NMD targets, which are used to show that "UPF3A, in general, represses NMD in murine cells and tissues", to measure the NMD efficiency in murine ESCs and rib-muscle derived fibroblasts, and tissues, including the spleen, thymus, and liver. We further used additional 26 NMD targets to support our findings. All these 33 NMD targets are rigorously tested and validated with studies from other groups and ours (McIlwain et al. 2010, PNAS; Weischenfeldt et al. 2012, Genome Biol; Huang et al. 2011, Mol Cell; Li et al. 2015, EMBO J; Shum et al. 2016, Cell). qPCRs and semi-quantitative RT-PCRs on these 33 NMD targets all showed that UPF3A depletion has no/minimal roles in NMD. Our data, combining all previous study, strongly indicate that UPF3A is

dispensable in NMD when UPF3B is present. However, we agree with the reviewer that transcriptome analysis on control and *Upf3a* KO cells and tissues would give more information on cell and tissue specific NMD effect of UPF3A in mice. We definitely will include the transcriptomic information when we characterize the UPF3A functions in ESC/NSC self-renewal/differentiation.

2. UPF3A and UPF3B are two paralogues of yeast *Upf3*. UPF3B is a widely accepted NMD factor negatively regulating mRNA stability. It is better for the authors to use *Upf3b* as a functional positive control.

Response: We thank the reviewer for this comment. We agree with the reviewer that UPF3B is a widely accepted NMD activator. However, UPF3B is a well-recognized weak NMD effector, too (Chan et al. 2009, Nature Struct Mol Biol; Shum et al. 2016 Cell; Wallmeroth et al. 2022; Yi et al. 2022, EMBO J, etc.). Knockdown or knockout of UPF3B weakly increased a small portion of NMD targets. Thus, in qPCR or semi-RT-PCR assays, using UPF3B may not cause detectable variance on NMD target expression. In this way, we choose to use strong NMD effector SMG6 as the positive control for our qPCR and semi-RT-PCR assays. As shown in Fig 3, Supp Fig 5, Supp Fig 6A, Supp Fig 7 and Supp Fig 9, *Smg6* KO causes strong upregulations of NMD target gene transcripts, which provides a good control in our research to show the NMD deficiency.

3. Only NMD targets of *Atf4* and *Ddit3* were downregulated, but some of other NMD targets were upregulated in *Upf3a* KO spleen tissue from adult mouse. So the conclusion "*Upf3a* deficiency weakly inhibits NMD in multiple tissues from adult mice" is not appropriate.

Response: This comment is valid. Only NMD targets of *Atf4* and *Ddit3* were downregulated in UPF3A KO spleens, while expressions of most other NMD targets in spleen, liver, and thymus, remained unchanged or slightly upregulated. These data indicate that UPF3A, in general, have no/minimal effect on NMD. Thus, we have modified the section title to "UPF3A is dispensable for NMD in murine tissues".

Minor issues

1. In figure 1C, the authors used PCR to identify KO line. For better understanding, the authors might provide primers information in figure 1A.

Response: We thank the reviewer for this suggestion. We have added the primer localization information on revised Fig 1A. The primers' sequences are listed in Material and Methods part (Page 7, Line 182-185).

2. In McIlwain et al paper, the level of *Eif4a2* normal transcript was much higher than PTC-containing isoform. However, In figure 3A, the *Eif4a2* PTC-containing isoform is higher than the normal transcript. Please give the explanation for the difference.

Response: We thank the reviewer for this suggestion. We checked the original figures from McIlwain et al (PNAS, 2010). In the PNAS study, the expression level of PTC⁺ isoform of *Eif4a2* is lower than the normal isoform. In our study (for example, Fig 3), semi-quantitative RT-PCR results showed that level of PTC⁺ isoform of *Eif4a2* is higher than the normal isoform. Deletion of well-recognized NMD factor *Smg6* resulted in an increase of the PTC⁺ isoforms (for example, Fig 3), which indicated the semi-quantitative RT-PCR approach we used is reliable. Thus, we reason that PCR systems (for example, PCR kits) may affect the amplification efficiency of different isoforms.

March 14, 2023

RE: Life Science Alliance Manuscript #LSA-2022-01589-TR

Dr. Tangliang Li
Shandong University
State Key Laboratory of Microbial Technology
No. 72, Binhai Road
Qingdao, Shandong 266237
China

Dear Dr. Li,

Thank you for submitting your revised manuscript entitled "UPF3A is dispensable for nonsense-mediated mRNA decay in mouse pluripotent and somatic cells". We would be happy to publish your paper in Life Science Alliance pending final revisions necessary to meet our formatting guidelines.

- please address the remaining Reviewer 1's concerns
- please upload your manuscript text as an editable doc file
- please upload your supplementary figures as single files
- please consult our manuscript preparation guidelines <https://www.life-science-alliance.org/manuscript-prep> and make sure your manuscript sections are in the correct order

A. FINAL FILES:

B. MANUSCRIPT ORGANIZATION AND FORMATTING:

Sincerely,

Reviewer #1 (Comments to the Authors (Required)):

There was generally sufficient response to this reviewer's original comments and request to better acknowledge the work of Wallmeroth et al (2022) and Yi et al (2022). Notably, additional detail around experimental design and procedures, and grammatical edits were made. Only the following few concerns remain:

1. Perhaps my concerns related the sex of the mESCs and RDSCs were unclear. The main concern stems from the fact that all four of the mESCs are female and harbor two copies of the X-linked paralog UPF3B, while male-derived RDSCs only have one copy. While X chromosome inactivation during mouse development is outside of the this reviewer's expertise, it is anticipated that differences in UPF3B expression levels between the mESCs and RDSCs likely impact expression levels and, consequentially, interpretation of the data on targeting NMD substrates in these two cell types. The authors need to clearly acknowledge the difference with these cells and comment on whether the mouse ESCs possess one or two active X chromosomes?
2. The justification for not showing qPCR data for all selected NMD targets in all cell types was sound; however, this reasoning should be included somewhere in the text of the article.
3. All references to 'WB' in text should be changed to 'western blot'.

Reviewer #2 (Comments to the Authors (Required)):

The revised version of the study entitled "UPF3A is dispensable for nonsense-mediated mRNA decay in mouse pluripotent and somatic cells" by Chen, Shen et al. was tremendously strengthened by the additional experiments, restructured text passages and overall improved clarity of the key message. Particularly the novel characterization of the UPF3A-UPF3B antibody and the knockdown experiments serve as a solid foundation for the remaining experiments and provide more functional insight.

In summary, the authors have adequately addressed all of my concerns and I have no further major comments or requests. Therefore, I support the publication of the manuscript in its present form.
(This report is from Volker Böhm)

Reviewer #3 (Comments to the Authors (Required)):

No further questions

To all reviewers:

We thank all reviewers for their efforts on assessing our revised manuscript. We are grateful for the reviewers' support on publication of UPF3A's NMD roles in murine cells and tissues.

Specific responses to reviewers are discussed below in blue.

Reviewer #1 (Comments to the Authors (Required)):

There was generally sufficient response to this reviewer's original comments and request to better acknowledge the work of Wallmeroth et al (2022) and Yi et al (2022). Notably, additional detail around experimental design and procedures, and grammatical edits were made. Only the following few concerns remain:

Response: We thank the reviewer for his or her support on publication. We have revised the text according to the suggestions.

1. Perhaps my concerns related the sex of the mESCs and RDSCs were unclear. The main concern stems from the fact that all four of the mESCs are female and harbor two copies of the X-linked paralog UPF3B, while male-derived RDSCs only have one copy. While X chromosome inactivation during mouse development is outside of the this reviewer's expertise, it is anticipated that differences in UPF3B expression levels between the mESCs and RDSCs likely impact expression levels and, consequentially, interpretation of the data on targeting NMD substrates in these two cell types. The authors need to clearly acknowledge the difference with these cells and comment on whether the mouse ESCs possess one or two active X chromosomes?

Response: The comment raised by the reviewer is valid since *Upf3b* is located on mouse X chromosome. Two active copies of *Upf3b* exist in female ESCs. Since only one X chromosome is present in adult male mice, it is predictable that protein level of UPF3B is higher in female ESCs than RDSCs from adult male animals. Furthermore, we found UPF3A protein is reduced in RDSCs. Thus, we revised the text and wrote "We found that UPF3A and UPF3B protein expressions are higher in female ESCs than in male somatic cells (mouse rib muscle derived somatic cells) (Fig 1D), indicating that expressions of UPF3A and UPF3B proteins are developmentally regulated. Since undifferentiated female ESCs have two active *Upf3b* gene copies, while only one *Upf3b* gene locus is present in male RDSCs, the relative amount of UPF3B vs. UPF3A is reduced in male RDSCs" (Page 15, highlighted in red).

However, with our data, we could not deduce a suitable reason to support the hypothesis that UPF3B expression could affect NMD substrates between different cell types and tissues. In male mice with one UPF3B copy and two UPF3A copies, relative expressions of UPF3A and UPF3B are distinct among tissues and cells (Fig 1D and supp Fig 8B). we could not see any NMD inhibition effect in these UPF3A deficient cells and tissues. Thus,

we feel the NMD biology of UPF3A and UPF3B is far more complicated, which deserves further investigation with *Upf3a/Upf3b* single and double knockout primary human and mouse cells.

2. The justification for not showing qPCR data for all selected NMD targets in all cell types was sound; however, this reasoning should be included somewhere in the text of the article.

Response: We have followed the suggestion from the reviewer. In the revised manuscript, we added a sentence "Of note, the expressions of several NMD target genes, such as *Smad5* and *Cdh11* in livers, *Smad7*, *Snhg12* and *Mettl23* in thymuses, could not be quantified due to the technical reason that melting curves of PCR amplicons showed multiple peaks" (Page 12, highlighted in red).

3. All references to 'WB' in text should be changed to 'western blot'.

Response: Thanks for the comments. We have replaced all "WB" with "western blot" in the text.

Reviewer #2 (Comments to the Authors (Required)):

The revised version of the study entitled "UPF3A is dispensable for nonsense-mediated mRNA decay in mouse pluripotent and somatic cells" by Chen, Shen et al. was tremendously strengthened by the additional experiments, restructured text passages and overall improved clarity of the key message. Particularly the novel characterization of the UPF3A-UPF3B antibody and the knockdown experiments serve as a solid foundation for the remaining experiments and provide more functional insight.

In summary, the authors have adequately addressed all of my concerns and I have no further major comments or requests. Therefore, I support the publication of the manuscript in its present form.

(This report is from Volker Böhm)

Response: We thank a lot to Dr. Böhm for his great enthusiasm on our findings. His suggestions greatly improved the quality of our manuscript.

Reviewer #3 (Comments to the Authors (Required)):

No further questions

Response: We thank the reviewer for his or her support on publishing our findings.

March 20, 2023

RE: Life Science Alliance Manuscript #LSA-2022-01589-TRR

Dr. Tangliang Li
Shandong University
State Key Laboratory of Microbial Technology
No. 72, Binhai Road
Qingdao, Shandong 266237
China

Dear Dr. Li,

Thank you for submitting your Research Article entitled "UPF3A is dispensable for nonsense-mediated mRNA decay in mouse pluripotent and somatic cells". It is a pleasure to let you know that your manuscript is now accepted for publication in Life Science Alliance. Congratulations on this interesting work.

DISTRIBUTION OF MATERIALS:

Again, congratulations on a very nice paper. I hope you found the review process to be constructive and are pleased with how the manuscript was handled editorially. We look forward to future exciting submissions from your lab.

Sincerely,
